# En route to dynamic life processes by SNARE-mediated fusion of polymer and hybrid membranes

Lado Otrin [1✉], Agata Witkowska [2,6], Nika Marušič [3], Ziliang Zhao [4], Rafael B. Lira [4,7], Fotis L. Kyrilis [5], Farzad Hamdi [5], Ivan Ivanov [3], Reinhard Lipowsky [4], Panagiotis L. Kastritis [5], Rumiana Dimova [4], Kai Sundmacher [3], Reinhard Jahn [2] & Tanja Vidaković-Koch [1]

A variety of artificial cells springs from the functionalization of liposomes with proteins. However, these models suffer from low durability without repair and replenishment mechanisms, which can be partly addressed by replacing the lipids with polymers. Yet natural membranes are also dynamically remodeled in multiple cellular processes. Here, we show that synthetic amphiphile membranes also undergo fusion, mediated by the protein machinery for synaptic secretion. We integrated fusogenic SNAREs in polymer and hybrid vesicles and observed efficient membrane and content mixing. We determined bending rigidity and pore edge tension as key parameters for fusion and described its plausible progression through cryo-EM snapshots. These findings demonstrate that dynamic membrane phenomena can be reconstituted in synthetic materials, thereby providing new tools for the assembly of synthetic protocells.

[1] Electrochemical Energy Conversion, Max Planck Institute for Dynamics of Complex Technical Systems, Magdeburg, Germany. [2] Laboratory of Neurobiology, Max Planck Institute for Biophysical Chemistry, Göttingen, Germany. [3] Process Systems Engineering, Max Planck Institute for Dynamics of Complex Technical Systems, Magdeburg, Germany. [4] Department of Theory and Bio-Systems, Max Planck Institute of Colloids and Interfaces, Potsdam, Germany. [5] Interdisciplinary Research Center HALOmem & Institute of Biochemistry and Biotechnology, Martin Luther University Halle-Wittenberg, Biozentrum, Halle/Saale, Germany. [6] Present address: Department of Molecular Pharmacology and Cell Biology, Leibniz-Forschungsinstitut für Molekulare Pharmakologie (FMP), Berlin, Germany. [7] Present address: Moleculaire Biofysica, Zernike Instituut, Rijksuniversiteit Groningen, Groningen, Netherlands. ✉email: otrin@mpi-magdeburg.mpg.de

At the dawn of a new genesis, as envisioned and facilitated by humankind, artificial life in its current iterations appears to be vividly reminiscent of the zygotic stage of multicellular life—full of potential and of ever-increasing complexity. In analogy to natural cells, the current synthetic constructs (synthetic cells and organelles[1–6]) that are being assembled from molecular building blocks in a bottom-up fashion, are predominantly envisioned as enclosed structures made of phospholipids, while cytosolic and membrane proteins (MPs) endow a plethora of natural functionalities to the otherwise passive containers. Even though these mimics rapidly develop complexity and elegance, they remain short-lived without the natural mechanisms for repair and replacement. Unsurprisingly, a good amount of effort is invested in increasing their stability and performance by replacing natural parts with man-made ones. Hereby, protein optimization via extremophilic sources and mutations is ultimately constrained by the intrinsic fragility of biomolecules, while chemical mimicking of complex MPs such as the rotary engine ATP synthase, nearly perfected through millions of years of evolution, is currently beyond our reach. On the other hand, significant progress can be achieved by augmenting the common delimiter and an essential component of artificial cells and organelles. In recent years, lipid membranes were successfully replaced with synthetic polymers, which even enabled successful insertion of ATP-synthesizing apparatus and retention of MP activity[7,8]. This substitution led to extended functional lifetimes[9,10], by counteracting enzyme delipidation for instance[11], while providing enhanced chemical resistance and lower permeability[11–13]. Altogether, the biocompatibility of the amphiphile to the MP is still explored on a case-to-case basis but a general roadmap is beginning to crystalize. For instance, the rigidity of the historical polymer poly(butadiene)-poly(ethylene oxide) (PBd-PEO)[14] arrested the oxygen reduction by a proton pumping terminal oxidase in a purely polymeric environment, though blending with phospholipids resulted in prolonged activity[10]. The latter strategy of membrane "hybridization" has been proposed to alleviate the drawbacks of natural and synthetic building blocks[15] and such mixed membranes have been investigated in detail, e.g., with respect to phase separation[16]. More fluid hydrophobic blocks like poly(dimethylsiloxane) (PDMS) combined with other hydrophilic chains such as poly(2-methyloxazoline) (PMOXA) and poly(2-ethyloxazoline) (PEtOz) have alone facilitated the reconstitution of complex bioenergetic proteins, e.g., ATP synthase[8] or complex I[17], next to plentiful channels like aquaporin[18]. Thereby, membrane fluidity, which was found to scale with the length of the hydrophobic block[19], and the hydrophobic mismatch between the bilayer and the MP[20], i.e., the membrane thickness, were found to be crucial parameters for unrestricted enzyme activity. This said, polymer membranes are not by default less permeable than natural ones[21] and light-driven proton pumps have also been reconstituted in "frozen" amphiphiles based on polystyrene[22]. In fact, the mechanical properties of polymersomes vary greatly[23], whereby the expanded design freedom and tunability are the particular assets of synthetic intervention. Thus, the design of ABC-type triblocks has allowed for the directed insertion of MPs thanks to membrane asymmetry[24], while a variation of the phospholipid type in hybrid films enabled control of MP partitioning in raft-like domains due to varying fluidities[25].

The next developmental stage of synthetic constructs, beyond purely channel or enzyme properties, is associated with the reconstitution of dynamic processes. Natural membranes not only segregate biochemical reactions and serve as interfaces for MPs but are also constantly being remodeled by these MPs and the cytoskeleton in processes like proliferation, signaling, and motility. The fusion of vesicles, together with fission, constitute probably the most important membrane remodeling processes. This enables the trafficking of cargo in various secretory pathways as well as of MPs and lipids[26], the latter providing a mechanism for membrane expansion in eukaryotic growth scenarios. On a practical note, vesicle fusion also allows for the reconstitution of multiple MPs with controlled orientation—a critical point in the assembly of synthetic constructs that requires further attention[27,28]. So far, the fusion of polymer membranes has been rarely addressed. Only physicochemical triggers have been applied to block copolymer vesicles with limited efficiency[29–32]. Furthermore, the fascinating process of protein-mediated fusion remains to this day entirely unexplored in polymersomes.

To address this challenge, we designed a fusion platform based on graft copolymer PDMS-g-PEO membranes, next to hybrid polymer/lipid mixtures, functionalized with fusogenic SNARE proteins. The choice of the comb-type siloxane polymer was directed by its suitable properties with respect to MP reconstitution[7,11] and not least of all, its commercial availability. We explored the SNARE-mediated fusion of polymer and hybrid membranes via membrane mixing, as well as through the functional coupling between two respiratory enzymes, $bo_3$ oxidase and ATP synthase. To elucidate the reasons for successful and efficient fusion, we investigated the change of bending rigidity upon SNARE insertion and the pore opening dynamics. In addition, we observed key fusion intermediates via cryo-electron microscopy (cryo-EM), which led to further insights of the fusion process in this system.

## Results and discussion

**Correct orientation of SNAREs and low bending rigidity of the synthetic membranes as prerequisites for fusion.** First, we reconstituted minimal SNARE fusion machinery—full-length synaptobrevin (syb) and the acceptor complex comprising syntaxin, SNAP-25, and a C-terminal syb fragment (referred to as "ΔN complex")—into polymer and lipid/polymer hybrid vesicles, and in parallel into liposomes, which served as a natural benchmark. To achieve this, we employed the so-called co-micellization method (for details see Methods) by using mixed micelles of SNAREs and amphiphiles, with sodium cholate as the mediating detergent. SNARE-functionalized nano-sized proteovesicles (Figs. S1, S2) were spontaneously formed upon detergent removal via size exclusion chromatography. Bearing in mind that only SNAREs with a correct (outward) orientation at the membrane will contribute to membrane fusion, we next analyzed their alignment via proteolytic digestion in the absence or presence of detergent. Analysis of SNARE fragments by SDS-PAGE (Fig. S3) revealed syb insertion with at least 81% and ΔN complex with over 92% correct orientations in all types of examined membranes, with the best overall orientation achieved in polymer vesicles (Fig. 1a). A similar trend of a more uniform outward orientation of the ΔN complex was observed previously in lipid vesicles[33]. Additionally, it is worth noting that only outwards-facing SNAREs could be observed via cryo-EM (Fig. S2). Furthermore, we assessed the stability and integration efficiency of inserted SNAREs with a flotation assay (Fig. S4). In this approach, proteovesicles are separated from the poorly incorporated/unstable SNAREs on a density gradient via ultracentrifugation. We observed stable and efficient integration of SNAREs in all types of membranes with marginal protein loss upon ultracentrifugation across all systems (Fig. 1b).

Following the successful and stable incorporation of SNAREs with predominantly correct orientation in the synthetic membranes, we next placed the proteovesicles in an environment that kept the system minimal, well-defined, and versatile. Towards this end, we used (unless specifically stated otherwise) a moderately

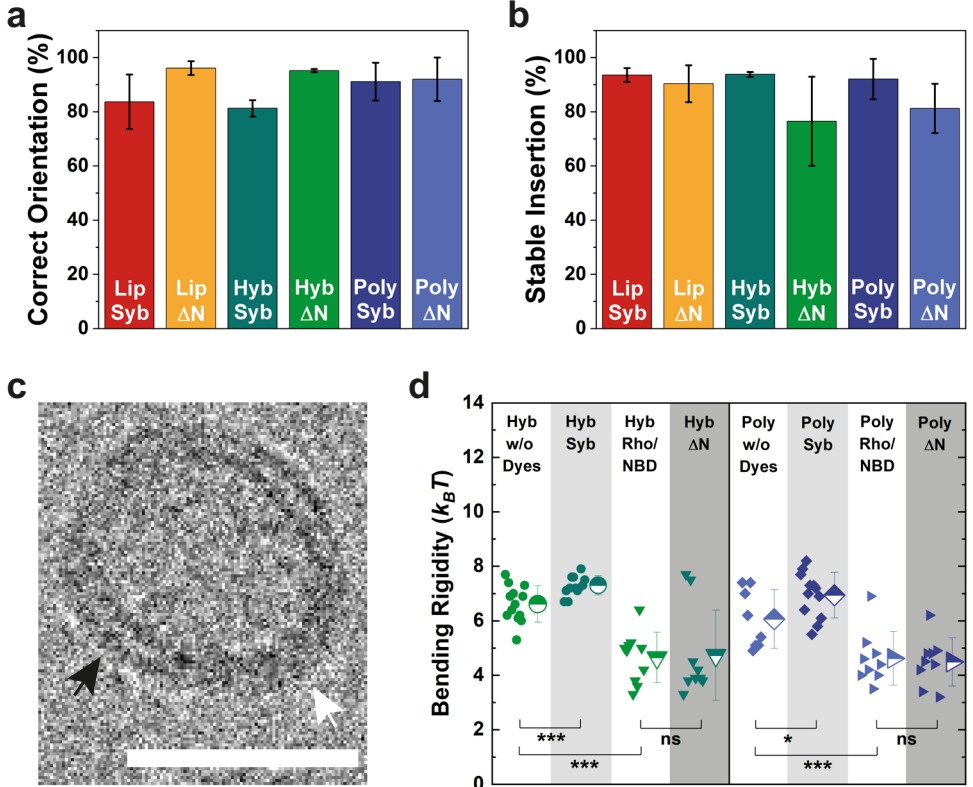

**Fig. 1 Characterization of SNARE-free and SNARE-functionalized vesicles. a** Fraction of outwards-facing SNAREs (Syb, synaptobrevin and ΔN, ΔN complex) in lipid (Lip), polymer (Poly), and hybrid (Hyb) vesicles determined by proteolytic digestion and fragment analysis via SDS-PAGE, and **b** fraction of firmly inserted SNAREs determined by a flotation assay on a density gradient. Mean values and standard deviations of at least two separate reconstitutions are shown. **c** Cryo-EM micrograph of a hybrid vesicle. Black and white arrows indicate lipid- and polymer-rich domains, respectively. Scale bar = 30 nm. **d** Changes in the bending rigidity of polymer and hybrid GUVs in 5 mM HEPES (pH 7.4), 5 mM KCl, and 30 mM sucrose upon dye (1.5 mol% each) and SNARE insertion determined by flickering spectroscopy. Shown are individual measurements as well as their mean values with standard deviations. ns, not significant for $P > 0.05$; *$P \leq 0.05$; ***$P \leq 0.001$.

buffered solution (20 mM) containing only HEPES, KCl, and dithiothreitol (DTT), at physiological pH, osmolarity, and ionic strength. We selected KCl since it is known for its relatively weak interactions with phospholipids[34] as well as with PEO[35]. This enabled a more accurate assessment of SNAREs as the predominant fusion mediator across all tested membranes by minimizing the contribution of ions/agitation towards membrane fusion. Since KCl was previously shown to induce membrane swelling[35], we probed for this effect by cryo-EM (Fig. 1c and Fig. S2) and measured the membrane thickness of polymersomes ($N = 578$) as well as of the polymer (Fig. 1c, single fuzzy contour; $N = 976$) and lipid (Fig. 1c, two parallel contours; $N = 171$) nanodomains of hybrid vesicles, with and without SNAREs. We determined a fairly constant thickness of PDMS-g-PEO in polymersomes ($6.6 \pm 0.6$ nm) and in hybrids ($6.1 \pm 0.3$ nm), while the lipid domains of the latter were on average $4.9 \pm 0.3$ nm thick. Interestingly, compared to vesicles formed in sucrose solution with low buffering capacity[11], the polymer swelled by nearly 20% (from $5.3 \pm 0.2$ nm), while the thickness of lipid bilayers remained unchanged. Furthermore, we observed no significant thickness change in any of the membrane compositions upon SNAREs insertion.

The presence of salts[36,37] and the reconstitution of MPs[11,38–40] can result in considerable changes of the membrane bending modulus $\kappa$—an important parameter governing the energy of a membrane and one of the several energetic barriers to fusion, with respect to the high curvature of the stalk[41]. To assess the effect of KCl, lipid dyes and inserted SNAREs on the bending

rigidity of polymer and hybrid vesicles, we scaled up the vesicle size to the micron range via fusion/electroformation. With this method, giant unilamellar vesicles (GUVs) were formed in 5 mM HEPES (pH 7.4) containing 5 mM KCl and their $\kappa$ was determined via fluctuation (flickering) spectroscopy[42]. Even though we had to considerably lower the salt concentration (compared to the membrane mixing experiments) in order to facilitate the formation of GUVs, this setup enabled us to gain valuable information on the influence of the aforementioned factors. Indeed, we saw a nearly 50% decrease of the bending rigidity in protein-free polymersomes and hybrids formed in KCl (Fig. 1d), compared to the ones grown in sucrose[11] (from $11.7 \pm 2.1$ $k_BT$ determined previously to $6.1 \pm 1.1$, and from $11.6 \pm 2.4$ to $6.6 \pm 0.7$, respectively). This observation is consistent with the previously reported effect of salts on neutral and charged lipid membranes[36,37] in conjunction with the proposed polymer loosening effect upon membrane insertion of various species[11]. The addition of the lipid dyes DOPE-N-(lissamine rhodamine B sulfonyl) (Rho) and DPPE-N-(7-nitro-2-1,3-benzoxadiazol-4-yl) (NBD) to polymer and hybrid vesicles led to further substantial softening. This was likely due to increased salt screening via the additional membrane charge introduced by Rho and NBD[37,43]. In contrast, the insertion of syb resulted in an only a moderate increase of the bending modulus in both polymersomes and hybrids, while the ΔN complex had no statistically significant effect. This outcome does not correspond to our previous findings[11] on membrane softening upon insertion of large transmembrane and multidomain proteins (ATP synthase and

$bo_3$ oxidase), likely reflecting considerable differences in protein size, architecture, charge, and concentration. In fact, a similar increase in the bending rigidity was previously reported for DNA, anchored in lipid membranes[44], a system that is more reminiscent of our current setup.

Lastly, KCl enabled us to neutralize the charged membrane dyes that were introduced to the membranes (discussed separately in the following sections), which allowed us to observe the fusion of artificial membranes in their near-native state with respect to surface charge.

**SNAREs prompt efficient membrane mixing of polymer and hybrid membranes.** Next, we examined whether SNAREs are able to induce membrane mixing in synthetic membranes. Towards this end, we supplemented one population of each type of vesicles with the fluorescence resonance energy transfer (FRET) couple Rho/NBD and reconstituted the ΔN complex. Meanwhile, we inserted syb in vesicles containing no dyes. With this setup (Fig. S5), membrane mixing can be followed by the dequenching of NBD upon fusion, which results from the dilution of labeled proteovesicles with unlabeled ones. Thereby, KCl neutralized the additional surface charge from Rho/NBD and thus minimized the electrostatic repulsion between the vesicles (Figs. 2a and S6). In control experiments, we omitted SNAREs from the vesicles, while keeping everything else the same in order to assess the sole contribution of KCl under mild agitation.

Compared to proteoliposomes, a very similar initial fast stage of membrane mixing was observed in proteopolymersomes and proteohybrids (Fig. 2b–d). However, while the NBD fluorescence in proteoliposomes reached a plateau after about 2 h, we measured a progressive increase at a slower but steady rate in both polymer-containing systems. This led to higher membrane mixing on average, whereby no saturation was seen over the duration of the experiment (Fig. S7). Some degree of membrane mixing was also observed in SNARE-free vesicles, albeit absent of the rapid initial stage, characteristic for SNAREs, which substantiated their predominant role as the fusion mediator. Interestingly, the overall faster membrane mixing in proteohybrids was seemingly not a result from superimposition of unmediated fusion. On the other side, the proteins appear to expedite the anticipated saturation as evidenced by the comparison between SNARE-functionalized and SNARE-free liposomes (Fig. 2b). We also complemented the fluorescence-based measurements with dynamic light scattering (DLS) measurements before and after fusion. Similarly, DLS showed an increase in vesicle size in all tested systems after 2 h (Fig. 2e–g). Finally, variations in the total membrane mixing within identical systems were previously reported for proteoliposomes[45]. We probed for this effect in proteopolymersomes and measured the variability over several separate reconstitutions in different hands, with values ranging from 18 to 24% after the 2 h benchmark, defined by the proteoliposomes (Fig. 2d, inset).

**Prolonged pore lifetime promotes content mixing.** The observed membrane mixing in polymer and hybrid vesicles provides strong evidence for the early steps of fusion, namely docking and hemifusion. However, membrane mixing does not allow to draw definite conclusions about pore opening and expansion, which are crucial during the later stages of fusion and enable the lumen of the two vesicles to merge. To gain insights into these later steps, we again used micron-scaled vesicles and porated them in electric fields to observe the closure dynamics of micron-sized pores. This technique has been applied at the dawn of polymersome research, and for PBd-PEO, the pore dynamics were correlated to the membrane thickness by microscopic

analysis[46]. Here, we expanded the analysis and quantified the pore edge tension[47]. Typical pore closure profiles of the three types of vesicles can be seen in Fig. 3a. The value of the edge tension for the lipid membranes used as a benchmark $16.4 \pm 7.4$ pN is in the lower range of values found for commonly used pure lipids like POPC ($25.8 \pm 6.4$ pN), whereas the edge tension in polymersomes ($7.9 \pm 4.2$ pN) and hybrids ($8.9 \pm 2.3$ pN) was even lower (Fig. 3b).

During electroporation, we noticed that several minutes after the electric pulse was applied and microscopic pores closed, some polymer vesicles lost contrast originating from the loss of sugar asymmetry across their membranes. To examine the porated vesicles more closely for the presence of persistent submicron pores, we electroporated the vesicles in the presence of the water-soluble and membrane-impermeable dye sulforhodamine B (SRB) to observe a potential leak in. We found that about a third of the polymersomes remained porated up to several minutes after the application of the pulse. A representative course of the dye entry can be seen in Fig. 3c but the dynamics varied greatly (Fig. 3d), which suggested that the defects differ in number and size. Finally, we investigated whether the submicron pores were resealed at later times by adding a second dye of similar size (ATTO 647) 5–10 min after electroporation. In all instances, in which the polymersomes were permeable to SRB, ATTO 647 was not observed in the lumen (Figs. 3e and S8). This indicates that the remaining submicron pores in the PDMS-$g$-PEO membranes eventually closed.

**Functional coupling as a crucial determinant for full fusion.** Next, we examined the content mixing of fused polymer and hybrid vesicles, primed by the SNARE-induced pore opening and expansion. We approached this by co-reconstituting one vesicle population, containing the ΔN complex with ATP synthase and another population, containing syb with $bo_3$ quinol oxidase (Fig. 4a). ATP synthase and $bo_3$ oxidase are respiratory enzymes involved in bacterial oxidative phosphorylation. The reduction of $bo_3$ oxidase via ubiquinol leads to translocation of protons across the membrane, establishing a proton gradient, which is then used by ATP synthase for the coupling of ADP and inorganic phosphate to ATP. In our setup, the reducing power is provided by DTT, while synthetic ubiquinone (UQ) 1 is used instead of bacterial UQ 8. The essence of this functional assay is that ATP synthesis can be achieved only when both enzymes are integrated into a shared compartment, thus enabling the bioenergetic coupling[27]. For a more reliable comparison, we carefully tailored the vesicle formation procedure so that all co-reconstituted vesicles described here would better match the ones employed in membrane mixing experiments (Figs. 4c–e and S9). Nevertheless, we had to make some changes in order to accommodate both respiratory enzymes, such as increasing the pH from 7.3 to 8.0 and lowering the temperature from 37 to 23 °C. We also supplied the necessary cofactors (phosphate, ADP) along with the reporter luciferin/luciferase system to enable functional testing. In this respect, we previously reported a decrease in the bending rigidity of liposomes, polymersomes, and hybrids upon insertion of $bo_3$ oxidase[11]. All mentioned changes could affect the early stages of vesicle fusion, manifested by membrane mixing, therefore the main focus here was solely on content mixing.

We combined the two populations of vesicles and, following short incubation, initiated proton pumping by $bo_3$ oxidase with the addition of DTT and UQ. As a result, we observed successful ATP production in both SNARE-functionalized and SNARE-free vesicles, although synthesis in the latter subsided rapidly (Fig. S10a–c), while the SNARE-mediated fusion resulted in a steady rate over a prolonged period of time (Fig. S10). The sole

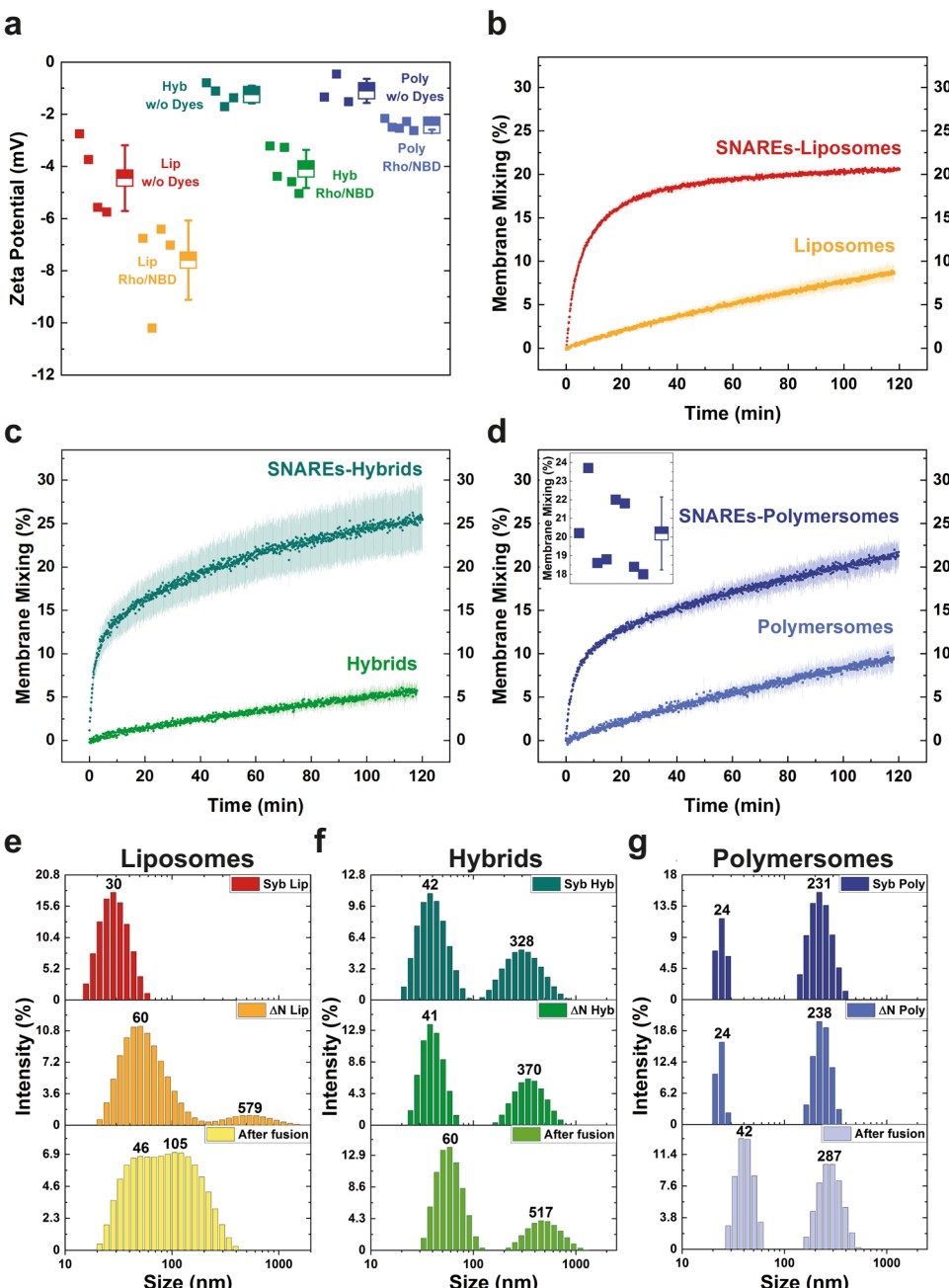

**Fig. 2 Membrane mixing upon SNARE-mediated fusion. a** The nearly neutral zeta potential of lipid (Lip), polymer (Poly), and hybrid (Hyb) vesicles promotes membrane mixing. The surface charge, introduced by lipid dyes was neutralized with KCl. **b–d** Comparison of SNARE-induced and spontaneous membrane mixing in different amphiphiles. Mean values of at least two different reconstitutions with standard deviations are shown. Inset of (**d**): Variability of SNARE-mediated membrane mixing in polymersomes. Individual measurements, as well as their mean value with standard deviation, are shown. **e–g** Intensity-based size distribution with indicated peak values of different vesicles upon synaptobrevin (Syb) or ΔN complex (ΔN) reconstitution before and after fusion. Size increase corresponding to several rounds of fusion can be seen in the lowest panels.

presence of various salts and cofactors, in combination with the energy provided by mechanical agitation, appears to suffice for some degree of fusion (consistent with the membrane mixing experiments). However, in the absence of guided mechanism, binary to the enzyme integration, this fusion is less efficient and random with respect to the functional coupling, which altogether leads to transient activity that cannot be sustained. The several-fold higher ATP synthesis rates achieved via SNARE-mediated fusion (Fig. 4b) further support this claim. Finally, we observed comparable ATP synthesis rates between SNARE-integrated liposomes, polymersomes, and hybrids, with polymersomes

exhibiting slightly higher activity than the other two platforms. However, this observation alone does not factor in the different protein integration efficiencies of both respiratory enzymes in these types of membranes. In fact, our previous findings[7] suggested that the integration efficiency of ATP synthase was the lowest in polymersomes and the highest in liposomes. Accounting for these differences by normalizing the ATP synthesis to the amount of reconstituted enzyme will lead to even better performance of polymer membranes. In either case it can be concluded that suitable membrane properties like the lower edge tension led to easier pore opening and expansion,

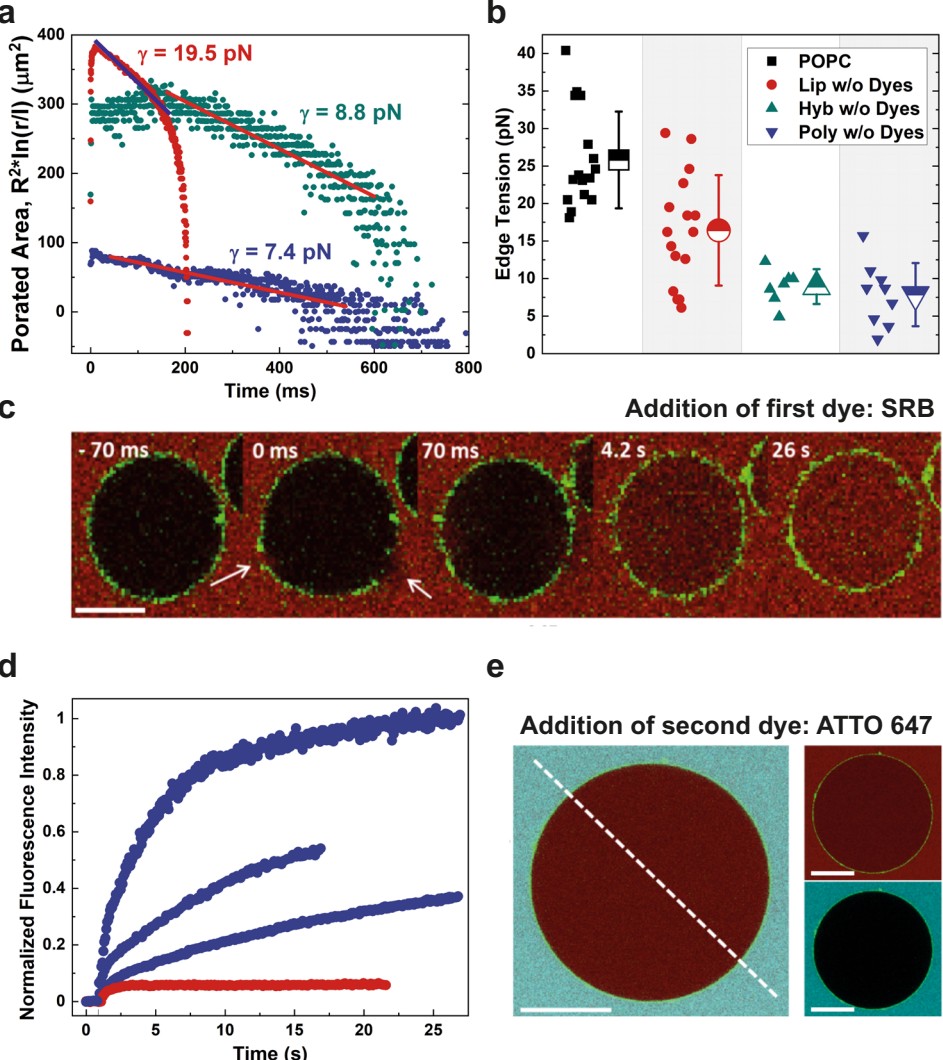

**Fig. 3 Pore opening dynamics in different vesicles. a** Typical pore closure dynamics of lipid (Lip; red), polymer (Poly; blue), and hybrid (Hyb; green) vesicles. Vesicle radius is denoted by $R$, pore radius by $r$, which is rescaled by the length $l = 1$ micron to avoid applying a logarithm to a dimensional parameter. The slopes of the data are used to determine the edge tension ($\gamma$) from the third stage of slow pore closure (blue line in the case of liposomes and red lines in the case of polymersomes and hybrids indicate linear regression). The pores in polymersomes and hybrids are more stable and remain open for longer times. The determined pore edge tension values are compiled in (**b**), each point representing a single GUV measurement along with the related mean values with standard deviations. **c** Time-lapse of SRB entry into polymer GUV after electroporation. Micron-sized pores (indicated by white arrows) close in the first ~100 ms after the pulse. Entry of dye was detected even after the closure of the large pores, indicating the presence of persistent submicron pores below optical resolution or out of the focal plane. Scale bar represents 30 µm. **d** SRB dye entry profiles of several polymer GUVs (blue) indicate differences in pore size, number, and lifetime. For comparison, typical dye entry profile of lipid GUV is depicted in red. **e** Absence of the second dye (ATTO 647 marked in cyan) in the vesicle lumen, added 5–10 min after SRB reveals closure of all pores. Fluorescence profiles of the depicted GUV along the indicated line can be found in Fig. S8. Scale bars represent 30 µm.

facilitating efficient SNARE-mediated fusion in both polymer and hybrid vesicles.

**Similar membrane composition but different fusion progression.** Lastly, with cryo-EM, we were able to capture fusion intermediates during the membrane and content mixing stages. Even though it is impossible to follow the process by destructive imaging, and the mixing of vesicle populations generates additional stochasticity, we reconstructed a plausible SNARE-mediated fusion progression in polymer and hybrid proteovesicles by comparison with natural systems[45] and molecular simulations[48,49]. Another argument for the postulated sequence was the fact that later intermediates were prevalent upon longer incubation times (Fig. S11).

We observed a docking stage of proteopolymersomes (Figs. 5-I and S12), which likely had proceeded towards the exvagination of a small membrane portion of a single vesicle, leading to the formation of a unilateral stalk (Fig. 5-II). Thus, a local point contact, which was also observed in some cases of liposome fusion[50], was established between the stalk and the proximal membrane (the act of "kissing"), which should, in turn, have initiated membrane mixing and led to the emergence of a narrow hemifusion diaphragm (Fig. 5-III). Considering the previously proposed organization of PDMS-*g*-PEO molecules (hairpin conformation) as a bilayer[51,52], we note that we adopted the terminology from lipid bilayers to provide an analogy, although some dissimilarities between the two systems cannot be excluded. Presumably, the diaphragm expanded further, and thinning of the

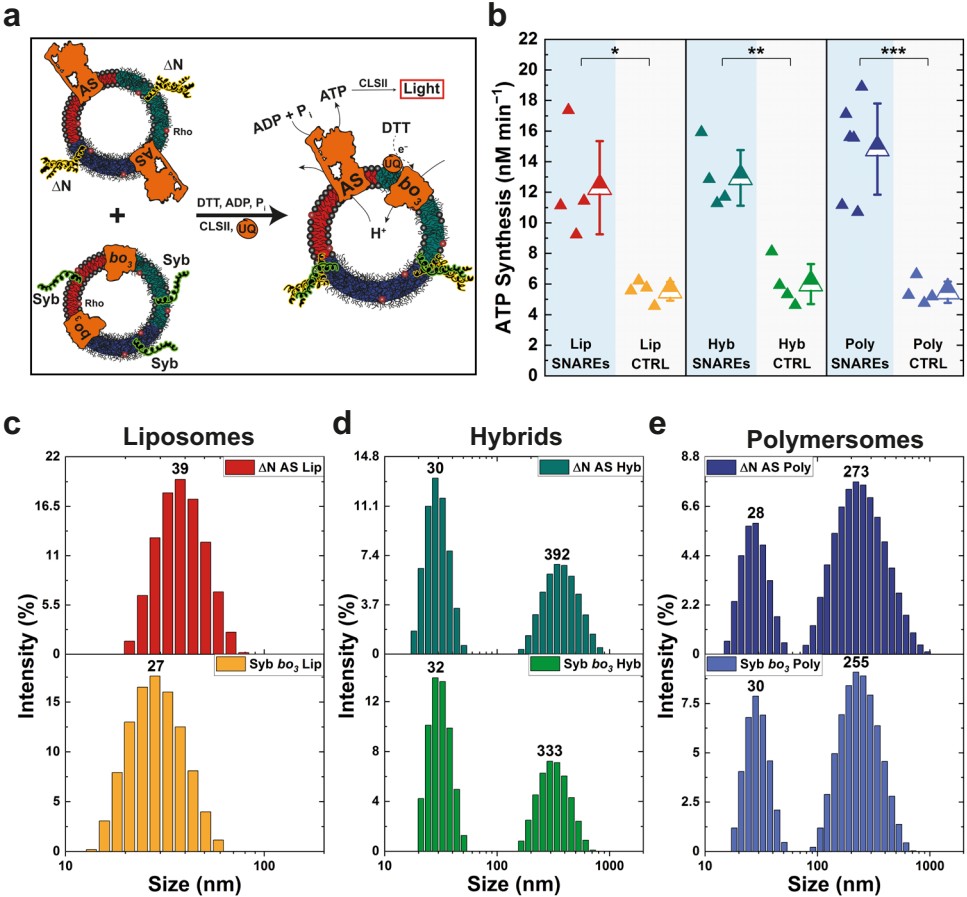

**Fig. 4 SNARE-induced content mixing in different vesicles. a** Content mixing was assessed via functional coupling between two enzymes of the bacterial respiratory chain: the proton pump $bo_3$ oxidase ($bo_3$) and the proton gradient consumer ATP synthase (AS). Each enzyme was co-reconstituted separately with either synaptobrevin (syb) or the ΔN complex (ΔN). Upon successful fusion, the gradient established upon activation of $bo_3$ oxidase with DTT was used by ATP synthase for ATP production. ATP was converted to luminescence signal via the luciferase/luciferin reporter system (CLSII). **b** Comparison of ATP synthesis rates after SNARE-free or SNARE-mediated fusion of different vesicles indicated higher coupling efficiency in the presence of SNAREs. Each point corresponds to a single measurement. Also shown are mean values with standard deviations. *$P \leq 0.05$; **$P \leq 0.01$; ***$P \leq 0.001$. The highest SNARE-mediated content mixing was achieved in polymersomes. **c–e** Intensity-based size distribution of different vesicles co-reconstituted with synaptobrevin (syb) + $bo_3$ oxidase ($bo_3$) or with the ΔN complex and ATP synthase (AS).

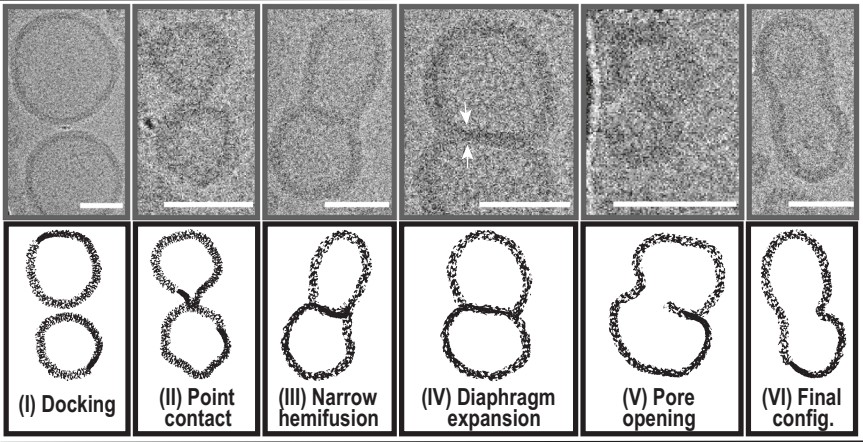

**Fig. 5 Proposed fusion intermediates of SNARE-mediated polymersome fusion.** Following vesicle docking (I; for an enlarged image please see Fig. S12) local point contact is established between the vesicles (II), leading towards membrane mixing and the emergence of hemifusion diaphragm (III). While the latter is expanding (IV), membrane thinning (shown with white arrows, enlarged in Fig. S13) can be observed at the juncture, indicating the location of the eventual pore opening (V). Such position of fusion pore was previously observed in simulations[49]. Slight lateral deformations in the newly fused vesicles along with less frequent inclusions in the lumen are likely remnants of the hemifusion diaphragm (VI). Scale bars represent 30 nm.

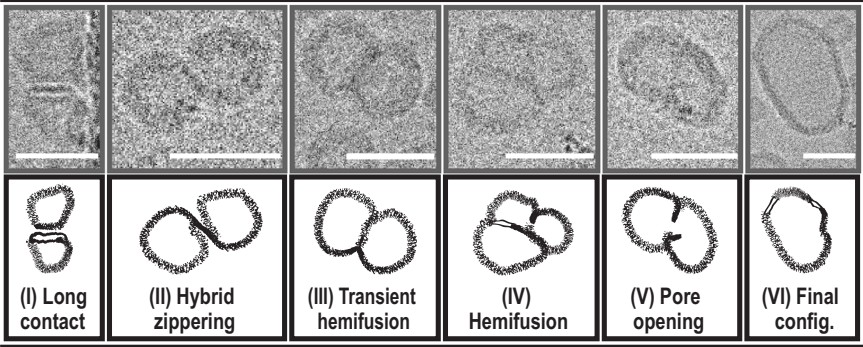

**Fig. 6 Proposed fusion intermediates of SNARE-mediated fusion in hybrids.** Long contact is established between the lipid-rich domain (bilayer) of one fusing vesicle, and the polymer-rich domain of another (I). The initial contact between the fusing membranes occurs at the lateral edge of the protruding bilayer and it further develops into crosswise membrane mixing (the process of "hybrid zippering", II). Transient fusion diaphragm, exhibiting lipid-rich bilayer stabilized in polymer bilayer is observed (III) before stable lipid bilayers are formed in the newly mixed hybrid diaphragm (IV). The fusion pore was formed in the more central region of the juncture (V), as previously observed in simulations[48]. No inclusions or deformations can be observed in fused vesicles (VI). Scale bars represent 30 nm.

membrane at the juncture, likely at the point of initial contact, was observed (Figs. 5-IV and S13). Furthermore, the pore opening (Figs. 5-V and S14) at the diaphragm extremity (previously observed in simulations[49]) suggested potential colocalization of the points of initial contact, membrane thinning, and pore opening. Finally, the resulting fused polymersomes featured persistent indentations, as well as occasional inclusions (bits of polymer membrane), likely remnants of the fusion diaphragm (Fig. 5-VI).

In contrast to proteopolymersomes, we discovered different steps of the fusion process in proteohybrids (Fig. 6). The docking stage in the latter was superseded by a long contact between the fusing vesicles, at a distance of about 2.5 nm. Remarkably, such contact was most frequently established between the polymer-rich domain of one vesicle and the lipid-rich domain (in a form of a bilayer) of another one (Fig. 6-I). Seemingly, the membrane mixing between the outer lipid layer and polymer bilayer originated at the lateral edge of the protruding lipid domain, and was propagated towards the opposite side (Figs. 6-II and S15). Such "hybrid membrane zippering" likely resulted in a state of transient hemifusion, in which lipid-rich bilayer domains (Fig. 6-II, III, exhibiting higher contrast compared to the polymer) were stabilized within the polymer bilayer. This temporary state was potentially resolved by membrane reorganization and the formation of lipid domains outside of the contact, leading to the formation of the characteristic configuration of hemifusion diaphragm, as in polymersomes (Fig. 6-IV). In all hybrid cases, we observed pore opening in a more central region of the diaphragm (Fig. 6-V), which continued with pore expansion, diaphragm dissolution, and, finally, membrane relaxation (Fig. 6-VI).

For mimicking natural membranes, we selected the graft copolymer PDMS-*g*-PEO, which previously enabled the insertion of complex MPs, while preserving their activity[7,9,11]. Furthermore, it is readily blended with lipids[7] to form macrohomogenous hybrid vesicles, offering a versatile environment for the reconstitution of MPs. Importantly, this polymer considerably extended the functional lifetime of integrated enzymes[9,11] and offered protection against oxidative damage[11]. We previously observed several PDMS-*g*-PEO characteristics of high potential to promote membrane fusion, in particular its low bending rigidity, sufficient fluidity, and comparatively low membrane thickness[7,11]. Here, we identified the low edge tension as another key material property that enabled easier pore opening and thus facilitated content mixing. Even though the structure and nanomechanics of

the nascent fusion pore of potentially proteolipid character[53] and the complex energy landscape of liposome fusion have not been fully resolved yet, there are indications that pore formation may be the dominating factor in the experimentally determined lumped activation energy of ~30 $k_B T$ in phosphatidylcholines[54]. In any case, it is safe to conclude that the joint contribution of easier bending and pore stability leads to a lower apparent activation energy for fusion in the polymer membrane compared to lipids. Thus, the 5 $k_B T$[55] of mechanical work, exercised by SNAREs at the conclusive stage of fusion will more easily overcome the energetic barrier and less cooperative effort would be required to achieve synaptic rates.

We believe that these advantages of PDMS-*g*-PEO are crucial merits in order to choose appropriate membranes for the construction of artificial cells and organelles, in particular with respect to membrane processes. The molecular basis for the beneficial properties of this particular polymer appears to be related to the flexible PDMS backbone and the grafted architecture, in contrast to the commonly used rigid diblocks based on PBd. In general, we do not expect that grafted arrangement will be the sole determinant of biomimicking since the mechanical properties can be tuned by variation of the hydrophilic/hydrophobic ratio and the molecular weight. On the other hand, high flexibility may present a trade-off and come at the expense of increased permeability, which would negatively affect bioenergetic, segregation, and signaling scenarios. However, this seems not to be an issue in the particular case since PDMS-*g*-PEO has similar water[56] and proton[11] permeability to lipids, while providing the emergent benefit of membrane resealing upon MP reconstitution in the case of mixed membranes. As stated above, the hospitality to MPs has to be ultimately proved in each specific case, while quantitative efforts for optimization of reconstitution protocols have already been undertaken[57]. Based on the current experimental evidences though, the commercially available PDMS-*g*-PEO accommodates optimal properties for the replacement of lipid membranes and further systematic studies may identify a general molecular roadmap for the design of nature-like or superior membranes.

The minimal fusion environment in the present setup was defined with as wide as possible biocompatibility in mind. Apart from HEPES, and DTT to reduce the SNAREs, the only other additive was KCl, which is commonly used to stabilize MPs and was chosen due to its comparatively weaker interaction with membranes. Thereby, KCl neutralized the surface charges introduced by lipid dyes, which minimized the electrostatic

repulsion between vesicles, and additionally stimulated fusion by lowering the bending rigidity of polymersomes and hybrids. The latter effects led to measurable events of unmediated fusion, both in content and membrane mixing experiments, but the significantly slower initial kinetics of the latter do not lessen the orthogonality of the SNARE-based approach. Remarkably, we observed fusion of polymersomes and hybrids with identical or better efficiency than in lipid vesicles. This is particularly exciting with respect to the previously reported decrease in the functional integration of ATP synthase with increasing polymer proportion[7,11]. In fact, we demonstrated that the current strategy can be employed as a useful practical tool for the integration of bioenergetic apparatus in all tested amphiphiles, paving the way to further applications for protein trafficking in natural and synthetic membranes. Furthermore, the biological congruity of SNAREs allows for immediate employment of natural NSF/SNAP recycling machinery to enable directed and sustained fusion, in contrast to one-off physicochemical triggers such as electrostatic attraction, bulk control by osmotic pressure, or biomimetic strategies like DNA-mediated fusion[58]. Such a virtually inexhaustible process will be particularly relevant for recursive phenomena like the growth of membrane in the context of proliferation.

Finally, in line with the demonstrated biocompatibility, protective properties, and enhanced fusion ability of PDMS-*g*-PEO, alongside the established role of polymersomes in drug delivery, SNARE mediation opens new ways for targeted administration of encapsulated small molecules and biologicals. Though in addition to the above characteristics, a suitable polymer needs to be cellularly degradable and/or easily eliminated. Having secured that, the prime synaptic arena of SNAREs instantly motivates exemplary scenarios for the symptomatic treatment of neurological disorders like Parkinson's disease; such synthetic synaptic vesicles would directly (re-)activate dopaminergic neurons, while preventing systemic clearance. Moreover, the demonstrated orthogonality of PDMS-*g*-PEO to the respiratory enzymes suggests that further functionalization of materials with matching mechanical properties via targeting peptides and antibodies (for instance to cross the blood–brain barrier and thus ease the administration) is attainable as well. Yet the role of SNAREs is much more diverse and we hope that the current findings, which demonstrate that similar principles of protein-mediated fusion apply for polymer membranes as well, will stimulate multiple applications in synthetic biology and biomedicine.

## Methods

**Purification of SNAREs**. Syb-2 (1–116)[59], truncated syb-2 (49–96)[60], syntaxin-1A (183–288)[33], and SNAP-25a (1–206)[61] all originating from *Rattus norvegicus* were overexpressed in *E. coli* strain BL21(DE3) and purified as described previously[62,63] with Ni$^{2+}$-NTA affinity followed by ion-exchange chromatography on Äkta system (GE Healthcare). All proteins with the exception of truncated syb-2 were purified in the presence of stabilizing detergents Na-cholate (Merck, 206986-87-0: C6445) and CHAPS (Merck, 331717-45-4: C5070). The ΔN complex comprising truncated syb-2, syntaxin-1A, and SNAP-25a was preassembled as described previously[60,64] by mixing mentioned constituent at the 1.5:1:1 molar ratio, respectively. Said mixture was then incubated at 4 °C overnight and purified by ion-exchange chromatography in the presence of CHAPS on the following day. This ΔN complex is structured in such a way so that the syb binding to the free site at the N-terminus is accelerated, resulting in faster fusion as well as preventing the formation of the dead-end 2:1 syntaxin:SNAP-25 complexes[60,64].

**Purification of ATP synthase and *bo₃* quinol oxidase**. The *E. coli* F$_1$F$_O$ ATP synthase was overexpressed in *E. coli* strain DK8(ΔuncBEFHAGDC) and purified with the Ni$^{2+}$-NTA affinity chromatography as described previously[65] in the presence of octyl glucoside (Glycon Biochemicals, D97001-C), Na-deoxycholate (Merck, 302-95-4: 30970), and sodium Na-cholate (Merck, 206986-87-0: C6445). The *E. coli bo₃* quinol oxidase cloned in plasmid pETcyo was overexpressed in *E. coli* strain C43(DE3)(ΔcyoABCDE) and purified as described by[66] in the presence of dodecyl maltoside (Merck, 69227-93-6: D4641).

**Preparation of SNARE-inserted nano-sized lipid, polymer, and hybrid vesicles with the co-micellization method**. Proteovesicles were formed from mixed micelles upon detergent removal, loosely following the integration strategy described previously[60,64], with considerable modifications.

Soy phosphatidylcholine (Avanti, 95%, 441601), polymer PDMS-*g*-PEO (Dow, DOWSIL™ OFX-5329 Fluid), or hybrid mixture (polymer:lipid = 7:3, mol:mol—this composition produces homogeneous, well-mixed hybrid membranes[7]), all dissolved in chloroform:methanol (2:1, V:V), were deposited into a round-bottom glass vial.

For the membrane mixing experiments, in addition to the dye-free vesicles, one population of vesicles was supplemented with 1.5 mol% of 1,2-dioleoyl-*sn*-glycero-3-phosphoethanolamine-*N*-(lissamine rhodamine B sulfonyl) (ammonium salt) —"Rho" (Avanti, 810150) and 1.5 mol% of 1,2-dipalmitoyl-*sn*-glycero-3-phosphoethanolamine-*N*-(7-nitro-2-1,3-benzoxadiazol-4-yl) (ammonium salt) —"NBD" (Avanti, 810144) and the lipid content (liposomes, hybrids) or polymer content was reduced proportionally. Both dyes were dissolved in chloroform:methanol (2:1, V:V), at 1 mg ml$^{-1}$.

For the content mixing experiments, vesicles were supplemented with 0.5 mol% Rho (Avanti, 810150), dissolved as described above, and the lipid content (liposomes, hybrids) or polymer content was reduced proportionally.

Then, the solvent was evaporated under a gentle stream of nitrogen and the produced thin films were further dried under nitrogen for 1.5 h.

Next, for the membrane mixing experiments, dry thin films were resuspended at the final concentration of 5 mM in 20 mM HEPES (Merck, 7365-45-9: H6147) (pH = 7.4/KOH), containing 150 mM KCl (Merck, 7447-40-7: P9541), 1 mM DTT (Merck, 3483-12-3: D9779), and 5% (m [g]/100 ml) Na-cholate, by rigorous vortexing (~1200 RPM) until all material was seen dislocated from the glass and homogenously resuspended. This yielded lipid/polymer/(dyes)/detergent mixed micelles. Next, to micelles containing Rho/NBD, ΔN complex was added at the lipid/polymer/hybrid mixture:ΔN complex ratio of 1000:1 (mol:mol) and to the dye-free micelles, syb was added at the lipid/polymer/hybrid mixture:syb ratio of 400:1 (mol:mol). Upon protein addition, reconstitution mixtures were mixed briefly with three short bursts (1000 RPM) and incubated at 23 °C for 5 min. Proteovesicles were then formed spontaneously upon the detergent removal via size exclusion chromatography on the PD Minitrap™ G-25 column (GE Healthcare) equilibrated with 20 mM HEPES (pH = 7.4/KOH), containing 150 mM KCl, 1 mM DTT. Eluted fractions containing lipid dyes were collected and pooled. Furthermore, volume-wise, corresponding eluted fractions containing dye-free vesicles were collected and pooled.

For the content mixing experiments, dry thin films were resuspended in 20 mM HEPES (pH = 8.0/KOH), containing 150 mM KCl, 1 mM DTT, 5% Na-Cholate as described above. To one population of mixed micelles, first, ΔN complex was added at the lipid/polymer/hybrid mixture:ΔN complex ratio of 1800:1, followed by mixing with three short bursts. Next, after 5 min, ATP synthase was added at the lipid/polymer/hybrid mixture:ATP synthase ratio of 81000:1 (mol:mol) followed by brief mixing, as before. To another population of mixed micelles, syb was added at the lipid/polymer/hybrid mixture:syb ratio of 500:1 (mol:mol), and *bo₃* oxidase at the ratio of 27000:1 with intermediate mixing steps, as described above. Reconstitution mixtures were then incubated at 23 °C for 5 min. Proteovesicles were then formed spontaneously upon the detergent removal via size exclusion chromatography on the PD Minitrap™ G-25 column equilibrated with 20 mM HEPES (pH = 8.0/KOH), containing 150 mM KCl, 1 mM DTT, and 40 mM KH$_2$PO$_4$ (Merck, 7778-77-0: P5655). All eluted fractions containing lipid dyes were collected and pooled.

For the cryo-EM imaging, vesicles were prepared in the exact same way as described for the membrane mixing experiments, except that the ΔN complex was reconstituted at the lipid/polymer/hybrid mixture:ΔN complex ratio of 1800:1 (mol:mol).

**Determination of the orientation of reconstituted SNAREs**. Orientation of SNARE proteins incorporated into liposomes, polymersomes, or hybrid vesicles was assessed with protease (trypsin) digestion of intact and detergent-solubilized vesicles as adapted from ref. [33]. Briefly, 40 µl of vesicles were incubated either with 10 µl of buffer only (20 mM HEPES pH 7.4 and 150 mM KCl), buffer and trypsin (Merck, 9002-07-7, final concentration 0.1 mg ml$^{-1}$), or buffer, trypsin and Triton X-100 (Merck, 9002-93-1, final concentration 0.3%). After 2 h incubation at 37 °C, samples were analysed by Tricine-SDS-PAGE[67]. With respect to this, the band intensity of samples, treated with buffer only (total amount of inserted SNAREs) was compared with the band intensities of trypsin-treated samples (amount of inwards-facing SNAREs) and trypsin/Triton-treated ones (amount of indigestible intramembrane fragments of SNAREs). The fraction of outwards-facing SNAREs was then calculated by subtracting the inwards-facing SNAREs and indigestible fragments from the total amount of inserted SNAREs and dividing it by the latter.

**Stability assessment of reconstituted SNAREs via the flotation assay**. Integration stability and efficiency was evaluated with a flotation assay on a discontinuous Nycodenz (Progen, 1002424) gradient, as described[33]. Briefly, reconstituted vesicles were mixed with an equal volume of 80% Nycodenz dissolved in 20 mM HEPES (pH 7.4/KOH), 150 mM KCl, and were overlaid first with 30% Nycodenz and then with said buffer only. Following ultracentrifugation,

reconstituted vesicles were partitioned in the uppermost (buffer) layer of the gradient, while the non-incorporated SNAREs as well as disassembled ΔN complex can be found in lower layers. The SNAREs content of different layers was analysed by Tricine-SDS-PAGE and the amount of SNAREs, inserted in a stable manner (upmost layer) was compared with the total amount of SNAREs in all layers.

**Zeta potential determination**. The Zeta potential of protein-free vesicles employed in the membrane and content mixing experiments was determined under experimental conditions with respect to solvent (buffer) composition, pH, temperature, and presence of various related cofactors (UQ, ATP, etc.). Towards this end, protein-free vesicles with the composition described above (with or without 1.5% of both Rho/NBD for membrane mixing and with 0.5% Rho for the content mixing) were prepared as described above in either membrane mixing buffer (20 mM HEPES (pH 7.4/KOH), 150 mM KCl, 1 mM DTT; viscosity at 37 °C = 0.760 cP, the refractive index at 37 °C = 1.3332) or in content mixing buffer (20 mM HEPES (pH 8.0/KOH), 150 mM KCl, 40 mM KH$_2$PO$_4$, 73.5 μM ADP, 58.7 nM ATP, 35.2 mM DTT, 0.6 mg ml$^{-1}$ CLSII luciferin/luciferase reagent, and 0.2 mM UQ; viscosity at 23 °C = 1.019 cP, the refractive index at 23 °C = 1.3364), respectively. In parallel, for a comparison, all vesicles were also prepared in milli-Q water. Prior to measurements, folded capillary zeta cells (Malvern Panalytical, DTS1070) were filled with either of two buffers or with milli-Q, and a small volume (10–40 μl) of vesicles, with the concentration of 80 mg ml$^{-1}$ was injected directly to the bottom of the cell. This so-called diffusion barrier technique minimized sample contact with the electrode and the resulting sample decomposition. Samples, intended for the content mixing were measured at 23 °C. Meanwhile, samples, intended for membrane mixing were deposited into capillary cells, incubated at 37 °C for 30 min prior to measurements, and then measured at 37 °C. All samples were measured with Zetasizer Nano ZS (Malvern Panalytical) using Smoluchowski approximation and monomodal analysis (i.e., fast field reversal) as well as general-purpose mode (combination of fast and slow field reversal) when possible, both yielding very similar results. Measurements consisted of 10–300 runs each. Reported are mean zeta potentials acquired in separate measurements.

**SNARE-mediated membrane mixing and vesicles size changes following fusion**. Vesicles intended for the membrane mixing experiments as described in section "Preparation of SNARE-inserted nano-sized lipid, polymer, and hybrid vesicles" were first incubated at 37 °C for 30 min following preparation to preheat them. Then, the vesicle population containing Rho/NBD was resuspended in preheated membrane mixing buffer (20 mM HEPES (pH 7.4/KOH), 150 mM KCl, and 1 mM DTT) in a stirred quartz cuvette. The NBD in tagged vesicles was excited at 460 nm and a baseline fluorescence emission of NBD at 535 nm was recorded on Varian Cary Eclipse (Agilent), with the ex./em. slits at the settings 10/10 and with the PMT voltage set at 480 V. Then, to initiate fusion, dye-free vesicles were added next. The final concentration of both vesicles was 0.4 mM each in a total reaction volume of 0.9 ml. Membrane mixing as an NBD dequenching was monitored until a plateau was reached. After that, 10% octyl glucoside resuspended in membrane mixing buffer was added stepwise (1–5 μl per addition) until maximal NBD dequenching was achieved. Typically, total NBD dequenching was reached at 0.53% final concentration of OG in liposomes, 0.43% in hybrids, and 0.48% in polymersomes. Measured changes in NBD emission were normalized using the initial NBD fluorescence as 0% and the fluorescence of max. dequenched NBD upon OG addition as 100%. In control experiments, SNAREs were omitted from vesicles. Reported are average values of at least two separate vesicle preparation with standard errors. Each point in Fig. 2a: inset represents separate vesicle reconstitution.

Intensity-based size distribution of vesicles before and after fusion was recorded with the Zetasizer Nano ZS at a fixed 173° scattering angle. Typically, three measurements were performed consisting of five runs with 70 s duration. The reported "before fusion" size estimation was recorded with the freshly prepared vesicles. Meanwhile, for the size estimation of vesicles after fusion, a small aliquot of the membrane mixing reaction mixture was removed after the reaction plateau was reached and prior to the addition of OG. After size was determined, measured volume was returned and OG was added.

**SNARE-mediated content mixing via coupling of respiratory enzymes**. Vesicles intended for the content mixing experiments were prepared as described in section "Preparation of SNARE-inserted nano-sized lipid, polymer, and hybrid vesicles" and their intensity-based size distribution was recorded as described in the previous section. To content mixing buffer (20 mM HEPES (pH 8.0/KOH), 150 mM KCl, 40 mM KH$_2$PO$_4$, 73.5 μM ADP, and 0.6 mg ml$^{-1}$ CLSII luciferin/luciferase reagent), both populations of vesicles were added with their final concentration of 2.5 mM each, and a baseline luminescence signal was recorded with GloMax® 20/20 luminometer (Promega) at 23 °C. Next, a known concentration of ATP (Merck, 34369-07-8: A7699) dissolved in milli-Q water was added to enable signal calibration. Proton pumping action by $bo_3$ quinol oxidase was initiated with the addition of (final concentrations) 35.2 mM of DTT dissolved in milli-Q and 0.2 mM of UQ 1, dissolved in DMSO, and a luminescence increase as a result of ATP synthesis by the ATP synthase was recorded. During measurements, the

samples were not stirred. The content mixing solutions were vortexed briefly every 5 min to enable oxygen supply and homogenous redistribution in measured volume. It is worth noting that the vesicles started fusing as soon as they were mixed, while there was a 5–10 min delay with the ATP synthesis detection so that the baseline/calibration could be recorded. The ATP synthesis rates were determined from the fast initial steady-state synthesis via linear regression, as depicted on Fig. S10b. Each data point in Fig. 4b represents separately reconstituted vesicles. A t-test (two-tailed, heteroscedastic) was conducted to determine the effects of SNAREs on the efficiency of content mixing.

**Bending rigidity determination with the flickering spectroscopy and related scale-up of nano-sized vesicles to micron range**. To enable the determination of bending rigidity of vesicles via flickering spectroscopy, first, we had to increase their size. Nano-sized vesicles with or without proteins/dyes were prepared as described in section "Preparation of SNARE-inserted nano-sized lipid, polymer, and hybrid vesicles" with the membrane mixing configuration (1.5% Rho/NBD, ΔN 1:1000, and syb 1:400) and with the starting concentration of polymer/hybrid films being relatively high, 17 mM. After size exclusion chromatography, said vesicles were diluted to about 5 mM, which we found to be optimal for the formation of GUVs. Micron-sized vesicles were formed with our previously established fusion/electroformation approach[11], which we further modified and optimized for SNAREs. First, vesicles at about 5 mM following their formation via size exclusion were deposited onto ITO-coated glass slides (55 Ω) as seven droplets, 2 μl each, followed by partial dehydration of deposited samples over the course of 30 min. Next, an electroformation chamber (consisting of two sandwiched ITO-coated glass slides separated by a 1.81-mm-thick silicone spacer) was assembled and filled with 5 mM HEPES (pH 7.4), 5 mM KCl, and 30 mM sucrose. Giant vesicles were formed by applying the following sequence in sinusoidal electric field: 500 Hz, 0.05, 0.1, 0.2, 0.3, 0.5, 0.7, 0.9, 1.1, 1.3, 1.5, 1.7, and 2.0 V for 6 min each; 500 Hz, 2 V for 20 h; and 4 Hz, 0.5 V for 30 min. When lipid dyes were used, the electroformation was performed in the dark.

For the fluctuation analysis, 60 μl of outer solution (5 mM HEPES (pH 7.4), 5 mM KCl, and 67 mM glucose; 85 mOsmol kg$^{-1}$) were deposited onto a coverslip (Knittel Glass, Thickness No. 1.5H (0.17 ± 0.005 mm), refractive index = 1.5255 ± 0.0015, 24 × 50 mm) of the observation chamber and 3–5 μl of formed GUVs (in 5 mM HEPES (pH 7.4), 5 mM KCl, and 30 mM sucrose; 48 mOsmol kg$^{-1}$) were pipetted directly into the droplet of outer solution and gently mixed with the pipette. Fluctuation analysis was performed following the protocol described earlier[42]. The data were acquired at 23 °C. The acquisition of 2400 snapshots was done by the high-resolution camera (pco.edge, PCO AG, Kelheim, Germany) with 200 μs exposure time and 20 fps frame rate (in phase-contrast mode and 40× (NA0.6) objective on inverted microscope Zeiss Observer.D1). Vesicle fluctuations were analysed using custom-built software as previously reported[42]. Vesicles containing inclusions, large buds or tubes, or ones that did not significantly fluctuate, were excluded from the analysis. A t-test (two-tailed, heteroscedastic) was conducted to determine the effects of dyes/SNAREs on the bending rigidity of polymer and hybrid vesicles.

**Determination of the pore edge tension of vesicles and the analysis of dye entry dynamics**. Protein-free GUVs were prepared using the previously described electroformation method[7]. In short, lipid, polymer, or the hybrid mixture of the two (7:3, mol:mol) dissolved in chloroform at the concentration of 1 mg ml$^{-1}$ was spread onto two ITO-coated conductive slides (about 25 μl total per slide) and the solvent was evaporated under a gentle stream of nitrogen over 1 h. Both slides were then connected via a 2-mm-thick Teflon spacer, which created a chamber with about 1.9 ml volume. Chamber was filled with 200 mM sucrose (Merck, 57-50-1: S7903) dissolved in milli-Q, and connected to a function generator. Vesicles were formed in the AC field of 1.5 V and 10 Hz at 23 °C over the course of 2 h. When lipid dyes were used, the electroformation was performed in the dark.

Membrane edge tension was measured according to the method reported in ref. [47] with some modifications, and using the theory of pore closure as previously developed[68].

In short, GUVs prepared as described above, but instead in 200 mM sucrose solution containing 0.1 mM NaCl (Merck, 7647-14-5: S7653) were dispersed in 180 mM glucose (Merck, 50-99-7: G7528) solution and observed under phase-contrast microscopy using an inverted microscope (Axiovert 135, Zeiss, Göttingen, Germany) equipped with a 40× (NA 0.6) Ph2 objective. Images were recorded using ultrahigh-speed camera Phantom v2512 (Vision Research, AMETEK) at 5 kfps. The obtained images were semiautomatically processed using ImageJ (NIH, USA) and analysed as described previously[47]. The slow, linear third stage of the pore closure was used in the analysis. For the electroporation, an electroporation chamber was formed by sticking two parallel copper strips (copper conductive tape, 1/4 inch width, 1101102, 3 M) onto a glass coverslip, separated by 0.5 cm. A closed chamber was formed by placing parafilm perpendicular to the strips, forming a chamber approx. volume of 100 μl. Prepared GUVs were diluted 10–20 times in isosmolar glucose solution and placed into the chamber for electroporation. The end of each tape was connected to a multiporator βtech pulse generator GHT_Bi500 (βtech, l'Union, France), and pulses of varying magnitude and time could be applied. If not mentioned otherwise, we applied a single DC pulse of 200 V and 4 ms duration. Since polymer GUVs often become permeable after the

application of an electric pulse (see Results), we exchange the GUVs in the chamber every time a pulse was applied and so vesicles were subjected to only one pulse.

The dye entry into electroporated GUVs was studied as in ref. [69] with minor modifications. Studied GUVs, labeled with 0.5 mol% of a green dye NBD were dispersed in a medium containing 2 μM of water-soluble dye sulforhodamine B (SRB)(3520-42-1: S1402), and were electroporated as described above. To test the long-term membrane permeability 5–10 min after the original poration, a second dye, ATTO 647 (ATTO-TEC) was added to the external solution of GUVs at the final concentration of 2 μM. Images were acquired using a Leica TCS SP5 confocal microscope (Wetzlar, Germany) using a 63x water-immersion objective (1.2 NA) at 512 × 512 pixels, 1 A.U., 400 Hz scanning speed, and three-line averages. To record fast electroporation dynamics in real time, videos were recorded with 128 × 128 pixels, at 1000 Hz scanning speed and one line average. NBD was excited at 488 nm using argon laser and its emission was detected at 495–550 nm. SRB and ATTO 647 were excited with a diode-pumped solid-state laser at 561 and 633 nm, respectively, and detected at 564–615 nm and 640–690 nm. As in above, the GUV samples were discarded after single electroporation.

**Cryo-electron microscopy**. The SNARE-inserted polymer and hybrid vesicles were prepared as described in the section "Preparation of SNARE-inserted nano-sized lipid, polymer, and hybrid vesicles". After the formation, first, vesicles were preheated at 37 °C for 30 min. Then, both populations of vesicles were mixed 1:1 (V:V) to initiate fusion and were being continuously stirred at 600 RPM at 37 °C. In the case of polymersomes, 8 min after the initiation, aliquots of fusing vesicles were collected and applied as 3.5 μl droplets onto a glow-discharged R2/1 type 200 Mesh Quantifoil holey carbon grid. Meanwhile, in the case of hybrid vesicles, aliquots of fusing vesicles were collected at 3 and 8 min after initiation to enable comparison between the potential difference in the observed frequency between early and late fusion intermediates. Samples were then vitrified using Vitrobot Mark IV System (Thermo Fisher Scientific) and standard Vitrobot Filter Paper (i. e., Ø55/20 mm, Grade 595) at 95% relative humidity and at 4 °C. A blot force of 2 and blotting time of 6 s were applied for vitrification. The grids with applied samples were mounted onto an FEI Glacios 200 kV autoloader under cryo conditions and were imaged with a Falcon 3EC direct electron detector in linear mode and a total dose of 50 e⁻/Å². Acquired movies were collected at a pixel size of 0.9612. Beam-induced motion correction was performed in RELION 3.0[70] using the built-in implementation of MotionCorr2[71].

Micrographs were analysed with ImageJ v1.53c software[72]. In the analysis of the frequency of fusion intermediates (Fig. S11), only elongated vesicles (Fig. 6-VI) and those featuring diaphragm remains were counted as "after fusion" states, likely leading to underestimation of the number of fully fused vesicles.

## Data availability
The authors declare that all the relevant data supporting the findings of the study are available in this article and its Supplementary Information file, or from the corresponding author (L.O.) upon reasonable request.

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

## Acknowledgements

The authors are grateful to Claudia Bednarz for her excellent support with the purification of MPs used in this study and to the group of Dr. Steffen Klamt (Analysis and Redesign of Biological Networks, Max Planck Institute for Dynamics of Complex Technical Systems) for access to their fluorescence spectrophotometer. Moreover, the authors acknowledge helpful comments by the students of Max Planck School Matter to Life. This work is part of the MaxSynBio consortium which is jointly funded by the Federal Ministry of Education and Research (BMBF; grant number 031A359A) of Germany and the Max Planck Society (grant number MIFA Bioc 8080). P.L.K., F.H., and F.L.K. acknowledge funding by DFG (project number 391498659, RTG 2467), BMBF (ZIK program, grant number 02Z22HN23) the European Regional Development Funds for Saxony-Anhalt (grant number EFRE: ZS/2016/04/78115), and Martin Luther University Halle-Wittenberg. R.J. acknowledges funding by the Max Planck Society and the US National Institutes of Health grant No. 2 P01 GM072694.

## Author contributions

L.O. and T.V.-K. conceptualized the study. L.O., A.W., N.M., Z.Z., R.B.L., F.L.K. and F.H. performed the experiments. L.O., A.W., N.M., Z.Z. and R.B.L. developed the methodology. L.O. and N.M. coordinated the research. R.L., P.L.K., K.S., R.D., R.J. and T.V.-K. provided funding and resources. All authors analyzed the data. L.O. visualized the data. I.I., R.L., P.L.K., R.D., K.S., R.J. and T.V.-K. supervised the research. L.O. and I.I. wrote the manuscript. All authors reviewed and edited the manuscript.

## Funding

## Competing interests

The authors declare no competing interests.
