## [Peer Review File · Nature Communications]

REVIEWER COMMENTS

Reviewer #1 (Remarks to the Author):

Comment on

« En route to dynamic life processes by SNARE-mediated fusion of polymer and hybrid membranes »

This work is part of the current trend aiming at reproducing elements of cell function via synthetic model systems. (So called bottom up approach or synthetic biology)

In this context the authors try to mimic cell fusion by functionalizing lipid/polymer and Hybrid vesicle with appropriated minimal SNARE fusion machinery and a simple mixing of the vesicles. The fusion is followed via DLS, coupling of respiratory enzymes, and the authors try to explain the mechanism of fusion by the mean of CryoTEM analysis on LUV/SUV, measurement of bending rigidity and pore-edge tension on GUV.

Globally the study is well conducted , however it suffers from a lack of explanation of the choice of the vesicular systems especially the choice of hybrid polymer/lipid vesicle. Apart from a previous study of the authors, the advantage of these system in such context is not underlined at all. The authors should describe more precisely the literature on this systems and why they are useful for such studies (which at the end is not so obvious regarding the results obtained)

I suggest that the authors consider the following remarks

End of the first paragraph of the introduction :

It is precised in the introduction the history of liposomes and their « replacement » by polymersomes in the context of membrane protein insertion. The authors also precise that fusion of polymer vesicle has been rarely considered, etc...but there is absolutely no mention of the context of hybrid vesicles. Why using such structure ? As previously mentioned the authors should enlarge the literature. A general comment that could be made on this literature section is the relation between hybrid membrane structure and protein insertion. Is there a relation (Presence of domains facilitate or not the membrane protein insertion ?). If so, what are the parameters which favour apparition of domains ?

The authors precise that polymers with a « careful design » replaced successfully lipid for MP insertion and retention of their activity. Ref 7-9 are cited.

The reference 7 corresponds to their own work, were they used commercial PDMS-g-PEG copolymer, which do not really present a controlled architecture (relatively polydisperse, number of PEG moieties on PDMS backbone is variable from one chain to another but in average, 2). However it is true that this copolymer does form vesicle very easily, with very thin membrane

The reference 8 corresponds also to the work of the authors with again the same Grafted copolymer and a PBut-b-PEO copolymer with a very low molar mass leading again to a very thin membrane.

The ref 9 is one of the pionner in the field of Insertion of MP in polymer membrane but restricted to triblock copolymer and one particular protein PR-GFP. But indeed this is more in the spirit of the « careful design » mention by the authors.

To my opinion the authors should reinforce this part, by precisising the tendencies extracted from literature (essentially gained from the work of Meier and col. (Extreme flexibility of the hydrophobic block necessary to adapt to protein size, work essentially conducted on PDMS based copolymers) this will give a reader a more general view on the problem.

Regarding the comment about lower permeability and corresponding reference 10,12,13) although it is indeed admitted in literature , quantitative values of permeability for polymersomes are scare

and polymersomes presenting high permeability has been already designed (See Battaglia¹ and Carlsen and al²) Especially the PDMS-g-PEO system used by the authors show permeability to water commonly observed on liposomes. The authors should modify this comment.

Paragraph : Preparation of snare inserted lipid.

It is precised that the composition used (Polymer/lipid 7/3 mol/mol) produces well mixed homogeneous hybrid membrane, but is mention later the existence of domain... (CryoTEM results...). The authors should clarify the choice of this composition.

Paragraph : Orientation of Snares :

The authors should indicate in the ESI the band intensities of the SDS Page analysis allowing the calculation of results indicated in figure 1. The authors should precise also in the text « the moderately buffered solution »

CryoTEM Fig 1 :

Black and white arrow indicates polymer and lipid rich domain...(fuzzy and two parallel contour) this is far to be obvious and it is not easy to understand this result regarding the choice of the composition which should produced well mixed homogenous membrane
The comment about the absence of variation of thickness upon membrane insertion is difficult to envisage : such variation should not be strong, and may be not detectable with so thin membrane and resolution of the cryoTEM.

Paragraph on Kb measurements

The authors want to evaluate flexibility of the membrane through the measurement of Kb and see influence of membrane protein insertion on GUV. However the insertion of MP and tests of fusion has been made on LUV/SUV. Can the results be transposed to what occurs for LUV /SUV ? What about membrane curvature effect ? Please add comments on this part.

Values of Kb are extremely Low (4-6kT) even lower that values reported for classical liposomes membranes (~15kT). Apart from the eventual effect of dye, snare etc were such values expected compared to pure vesicle (pure polymersomes, pure soy PC ? what about literature ?)

Paragraph :Fusion and light scattering experiment :

There is no comment about the smallest population in Hybrid vesicles (around 40nm which evaluate to 60nm after fusion) and 24nm for pure polymersomes. As it is intensity based size distribution, this is the major population. Is it correlated with CryoTEM results ? It would be nice for cryotem to add pictures with larger scale and precise size distribution by cryotem (on a few hundred of vesicles)

Is there a role of the preparation process on the fusion of the vesicle ?

Pure polymersomes vesicle fuse progressively during time which is unexpected. Are the Polymersomes prepared by classical procedure (hydration extrusion process) fuse during time ? This should be checked.

Paragraph : Functional coupling as crucial determinant for full fusion :

Finally, the polymersome alone present higher ATP synthesis than the other systems : is it because this polymersome is particular in the sense that it forms very thin membrane similar to those of liposomes and that it is very flexible ?

Is there a specific role of hybrid polymer/lipid vesicle here ?

Paragraph : Similar membrane composition but different fusion progression

It is mentioned that PDMS-g-PEO predominantly form monolayers, but it has been reported in previous studies on this copolymer, through Langmuir isotherm experiments and Static light scattering studies that this copolymer do form membrane with a bilayer (hairpin conformation of the chains).³ The authors should precise the reference where this information about predominant monolayer is available for this copolymer and/or discuss this section in regard of the paper aforementioned. May be mechanism of hemifusion can be applied to this copolymer.

Conclusion

In the conclusion the authors present the PDMS-g-PEO as ideal copolymer because of its high fluidity compared to the commonly used rigid diblock PBut-PEO. The copolymers PBut-b-PEO are not the most commonly used for membrane protein insertion but triblock PMOXA-PDMS-PMOXA or diblock PDMS-b-PMOXA are (See work of Palivan, Meier and col. for instance)⁴⁻⁷. Finally the authors should better precise in the conclusion what are the advantages/disadvantages of the particular grafted copolymer PDMS-g-PEO (apart to be commercially available). As aforementioned this system is relatively permeable, therefore would not be an ideal candidate in the development of nanoreactor/artificial cell well active control of diffusion of species is mandatory with low contribution of passive diffusion.

(1) Battaglia, G.; Ryan, A. J.; Tomas, S. Polymeric vesicle permeability: A facile chemical assay Langmuir 2006, 22, 4910.

(2) Carlsen, A.; Glaser, N.; Le Meins, J.-F.; Lecommandoux, S. Block Copolymer Vesicle Permeability Measured by Osmotic Swelling and Shrinking Langmuir 2011, 27, 4884.

(3) Dao, T. P. T.; Brûlet, A.; Fernandes, F.; Er-Rafik, M.; Ferji, K.; Schweins, R.; Chapel, J. P.; Fedorov, A.; Schmutz, M.; Prieto, M.; Sandre, O.; Le Meins, J. F. Mixing Block Copolymers with Phospholipids at the Nanoscale: From Hybrid Polymer/Lipid Wormlike Micelles to Vesicles Presenting Lipid Nanodomains Langmuir 2017, 33, 1705.

(4) Garni, M.; Einfalt, T.; Lomora, M.; Car, A.; Meier, W.; Palivan, C. G. Artificial Organelles: Reactions inside Protein-Polymer Supramolecular Assemblies Chimia (Aarau) 2016, 70, 424.

(5) Kowal, J. Ł.; Kowal, J. K.; Wu, D.; Stahlberg, H.; Palivan, C. G.; Meier, W. P. Functional surface engineering by nucleotide-modulated potassium channel insertion into polymer membranes attached to solid supports Biomaterials 2014, 35, 7286.

(6) Tanner, P.; Baumann, P.; Enea, R.; Onaca, O.; Palivan, C.; Meier, W. Polymeric Vesicles: From Drug Carriers to Nanoreactors and Artificial Organelles Accounts of Chemical Research 2011, 44, 1039.

(7) Kumar, M.; Grzelakowski, M.; Zilles, J.; Clark, M.; Meier, W. Highly permeable polymeric membranes based on the incorporation of the functional water channel protein Aquaporin Z Proc. Natl. Acad. Sci. 2007, 104, 20719.

Reviewer #2 (Remarks to the Author):

The review is attached.

Reviewer #3 (Remarks to the Author):

In their manuscript, Otrin et al. report the fusion of polymer-based membranes induced by the fusogenic SNARE proteins. The authors show that SNARE proteins can be incorporated with good orientation in polymer and hybrid lipid/polymer membranes, characterize the bending rigidity and pore edge tension of the resulting membranes, and devise a plausible fusion mechanism based on the observation of key fusion intermediates by cryo-electron microscopy.

This is to the best of my knowledge the first report showing that SNARE proteins can induce fusion of polymer-based membranes, which makes it an important contribution to the field of synthetic cells. Extending the use of natural fusogenic proteins to artificial and hybrid membranes opens indeed perspectives for the reconstitution of life-like processes in well-defined artificial systems. Therefore, I recommend publication in Nature Communications. However, a number of critical points should be addressed before publication.

Main comments

1/ As a general comment, I found that the manuscript lacks clarity in some places, mainly because the order in which results are introduced is sometimes confusing.

* In the first paragraph of the results section, the authors write that “SNARE-functionalized nano-sized proteovesicles were spontaneously formed upon detergent removal”, but no data is shown here to back-up this claim. The authors should already refer here to DLS studies and cryo-EM images to back up the fact that nano-sized proteovesicles were indeed formed (these results are only shown later in the manuscript).

* The section about membrane mixing is quite confusing to me. I would suggest rephrasing/re-organizing most of this section. Results are a bit mixed up: for instance, the authors first discuss the absence of signal saturation in hybrids and polymersomes, then they switch to DLS results, then discuss the kinetics of membrane mixing, then go back to discussing the saturation in liposomes... Please re-organize in a more logical way. Similarly, I don't get the logic in the first paragraph with the sentences “Thereby, KCl neutralized the additional surface charge ...” and then “In addition, we omitted SNAREs from the vesicles...”. This is confusing (e.g., should “In addition” actually be “In control experiments...”). I would remove the comment “which should be valid for the polymer systems as well but at later times” or provide data at longer times to back up this point. Last, I am not sure why the variability in membrane mixing is shown as an inset in Figure 2A. I think this should rather be shown in Figure 2D. Also, could the authors provide variability data on membrane mixing for the other two systems (liposomes and hybrids)?

* In the content mixing experiment, discussion of the results shown in Fig. S7 should be more accurate. See in particular the sentence “As a result, we observed successful ATP production in both SNARE-functionalized and SNARE-free vesicles, although synthesis in the latter subsided rapidly (Fig. S7A), while the SNARE-mediated fusion resulted in a steady rate over more than an hour (Fig. S7B).” Actually, Fig. S7A shows results for hybrid vesicles, while Fig. S7B shows results for polymersomes, so the two figures are about two different systems. This should be explicit in the sentence. Is there any evidence that SNARE-mediated fusion also gives a steady rate over more than an hour for the other systems (liposomes and hybrids)? Could the authors also provide representative examples of the time-dependent ATP synthesis (as shown in SI Figure S7 for hybrids) for all three systems, with and without SNAREs? Also, why the y-axis on Fig. S7 has not been converted to ATP concentration as in

Fig. S7A?

* Last comment about clarity issues: in a few places the authors compare their current results to previously published studies, without mentioning the values they obtained in these previous studies. For instance, they write: “compared to vesicles formed in sucrose solution with low buffering capacity, the polymer swelled by nearly 20%, while the thickness of lipid bilayers remained unchanged”. It may be clearer to quote these precedent values from ref. 10. Similarly, they write “we saw nearly 50% decrease of the bending rigidity in protein-free polymersomes and hybrids formed in KCl, compared to the ones grown in sucrose.” Again, hard to judge if this assertion is valid without reading the cited reference. Please provide any relevant value directly in the manuscript. Also, were studies in reference 10 performed on the same vesicle systems?

2/ Data should be added to back up the protein orientation studies. In particular, the corresponding SDS-PAGE gels should be shown in the SI. Also, there are no error bars on Fig. 1A, so it is unclear what the statistical relevance of these results is. Same comment about error bars applies to Fig. 1B.

3/ When the authors report membrane thickness values measured on cryo-EM images, what is the statistical relevance of the reported values? E.g., how many proteovesicles were analysed? Please provide any relevant statistics.

4/ Why are there two vesicle populations on DLS measurements (Fig. 2E-G)?

Other comments

5/ In the introduction, I reckon studies from Lecommandoux’s and Palivan’s groups on the construction of polymer vesicles as synthetic cells should be cited.

6/ I would suggest plotting Figs. 2B-D with the same scale on the y-axis, this would make it easier to compare the data. Same comment for Figs. 2E-G regarding the x-axis: easier to compare if the same scale on the x-axis is used (but the scale on the y-axis can be kept as it is now).

7/ Please define what CM stands for in SI Figure S6 (I guess it stands for “content mixing”?).

8/ The overall manuscript is a bit long (and in particular the conclusion), so might be worth condensing it a bit.

We would like to express our sincere gratitude to the Reviewers for the careful examination of our work. Their comments were of tremendous help to identify points of lesser clarity, all of which we addressed in this revision. Some sections were expanded to include additional discussion on polymer and hybrid vesicles or other requested data. Moreover, several new figures were added and some of the existing figures were revised according to the recommendations. Detailed responses to the Reviewers comments are included below.

REVIEWER 1

Comment on

« En route to dynamic life processes by SNARE-mediated fusion of polymer and hybrid membranes »
This work is part of the current trend aiming at reproducing elements of cell function via synthetic model systems. (So called bottom up approach or synthetic biology)

In this context the authors try to mimic cell fusion by functionalizing lipid/polymer and Hybrid vesicle with appropriated minimal SNARE fusion machinery and a simple mixing of the vesicles. The fusion is followed via DLS, coupling of respiratory enzymes, and the authors try to explain the mechanism of fusion by the mean of CryoTEM analysis on LUV/SUV, measurement of bending rigidity and pore-edge tension on GUV.

1.) Globally the study is well conducted , however it suffers from a lack of explanation of the choice of the vesicular systems especially the choice of hybrid polymer/lipid vesicle. Apart from a previous study of the authors, the advantage of these systems in such context is not underlined at all. The authors should describe more precisely the literature on this systems and why they are useful for such studies (which at the end is not so obvious regarding the results obtained)

The addition of lipids to polymer membranes (i.e., the formation of hybrid vesicles) is of particular importance for the insertion of membrane proteins, considering the essential role of lipids as facilitators of membrane protein (MP) activity. Cardiolipin, for instance, was shown to induce conformational changes in respiratory complex I by modulating the accessibility of the quinone to the enzyme (10.1126/sciadv.aav1850). The essential role of cardiolipin in promoting enzymatic activity was also observed in other respiratory enzymes, for example in cytochrome c oxidase (10.1007/BF00762857) or in complex III (10.1016/S0021-9258(19)69888-1, 10.1096/fj.02-0729fje), where the enzyme activity was reduced 28–50 % in the absence of this lipid. Similarly, in some cases when complex MPs were inserted into polymersomes comprising certain block co-polymers, no enzymatic activity was detected in the absence of lipids (10.1039/C6CC04207D). We speculate that this might have been the case also in 10.1021/nl051896e since lipids were effectively introduced in the form of purple membranes. Finally, we recently reported on several other beneficial properties of hybrid membranes, for instance membrane resealing after protein insertion, evidenced by the related decreased permeability to protons, along with the extended functional lifetime of reconstituted enzymes (10.1073/pnas.1919306117, 10.1002/cbic.201900774).

Taken into consideration that a variety of synthetic cells and organelles feature MPs (for instance the ones involved in energy regeneration) and that we explore SNAREs as tools for the assembly of such complex systems, we believe hybrid membrane architecture to be of significant importance. Due to this reason, we investigated SNARE-mediated fusion of hybrid vesicles along with the lipid and polymer ones.

Following the suggestions by Reviewer 1 as well as Reviewer 3, the Introduction and Conclusions sections were expanded in order to describe our motivation in greater detail. In this context, several relevant works by groups, working in a similar direction were discussed in the revised version of the manuscript.

I suggest that the authors consider the following remarks

2.) End of the first paragraph of the introduction : It is precised in the introduction the history of liposomes and their « replacement » by polymersomes in the context of membrane protein insertion. The authors also precise that fusion of polymer vesicle has been rarely considered, etc...but there is absolutely no mention of the context of hybrid vesicles. Why using such structure ? As previously mentioned the authors should enlarge the literature. A general comment that could be made on this literature section is the relation between hybrid membrane structure and protein insertion. Is there a relation (Presence of domains facilitate or not the membrane protein insertion ?). If so, what are the parameters which favour apparition of domains ?

Sections Introduction and Conclusions were expanded following the suggestions of Reviewer 1 and 3, as mentioned above. Furthermore, the protein insertion in relation to certain membrane properties was discussed briefly, as recommended. While in this work, we have not observed preferred insertion of SNAREs into either lipid or polymer phase in the hybrid membranes, this cannot be generalized and should be examined for each individual enzyme and hybrid membrane. In general, one could reasonably expect different protein partitioning in cases, where the two (or more) phases differ significantly (e.g., different membrane thickness and the related hydrophobic mismatch), while in the present case, both phases were very similar.

3.) The authors precise that polymers with a « careful design » replaced successfully lipid for MP insertion and retention of their activity. Ref 7-9 are cited.

The reference 7 corresponds to their own work, were they used commercial PDMS-g-PEG copolymer, which do not really present a controlled architecture (relatively polydisperse, number of PEG moieties on PDMS backbone is variable from one chain to another but in average, 2). However it is true that this copolymer does form vesicle very easily, with very thin membrane

The reference 8 corresponds also to the work of the authors with again the same Grafted copolymer and a PBut-b-PEO copolymer with a very low molar mass leading again to a very thin membrane.

The ref 9 is one of the pioneer in the field of Insertion of MP in polymer membrane but restricted to triblock copolymer and one particular protein PR-GFP. But indeed this is more in the spirit of the « careful design » mention by the authors.

To my opinion the authors should reinforce this part, by precisising the tendencies extracted from literature (essentially gained from the work of Meier and col. (Extreme flexibility of the hydrophobic block necessary to adapt to protein size, work essentially conducted on PDMS based copolymers) this will give a reader a more general view on the problem.

Please see Reviewer 1, Points 1 and 2. These comments were taken into account when the segments were expanded. We would like to express again gratitude for the in-depth literature analysis and valuable suggestions. "Careful design" here was related to the construction of the entirety of the proposed fusion platform and not to polymer design or synthesis. This segment was rewritten to improve clarity.

4.) Regarding the comment about lower permeability and corresponding reference 10,12,13) although it is indeed admitted in literature , quantitative values of permeability for polymersomes are scarce and polymersomes presenting high permeability has been already designed (See Battaglia1 and Carlsen and al2) Especially the PDMS-g-PEO system used by the authors show permeability to water commonly observed on liposomes. The authors should modify this comment.

This corresponding segments were modified as suggested to better reflect the determined permeability of PDMS-g-PEO in the context of polymersome permeability in general.

5.) Paragraph : Preparation of snare inserted lipid. It is precised that the composition used (Polymer/lipid 7/3 mol/mol) produces well mixed homogeneous hybrid membrane, but is mention

later the existence of domain... (CryoTEM results...). The authors should clarify the choice of this composition.

The description mentioned by Reviewer 1 refers to the absence of visible micro-domains, as reported previously (10.1021/acs.nanolett.7b03093). Nevertheless, perfectly homogeneous distribution of lipids and polymer in hybrid membranes can by no means be expected and domains can be observed on a nano-scale (as reported in this work). Phase separation (on a micro-scale) was previously reported (10.1039/C2SM07188F) to result in domain budding, leading to disintegration of hybrids into pure polymersomes and liposomes. Meanwhile no such effect was observed in homogenous hybrid membranes and for that reason, the latter were chosen in this work to ensure that the system remained stable and well-defined.

6.) Paragraph : Orientation of Snares :

The authors should indicate in the ESI the band intensities of the SDS Page analysis allowing the calculation of results indicated in figure 1. The authors should precise also in the text « the moderately buffered solution »

Pictures of SDS-PAGE gels and the related band intensities were added to SI (new Fig. S3, Fig. S4 were created). Furthermore, buffer strength was specified in the text, as suggested.

7.) CryoTEM Fig 1 :

Black and white arrow indicates polymer and lipid rich domain...(fuzzy and two parallel contour) this is far to be obvious and it is not easy to understand this result regarding the choice of the composition which should produced well mixed homogenous membrane

The comment about the absence of variation of thickness upon membrane insertion is difficult to envisage : such variation should not be strong, and may be not detectable with so thin membrane and resolution of the cryoTEM.

Please see Reviewer 1 Point 5 for comments on the homogeneity of the membrane. Regarding the detection of changes in membrane thickness upon protein insertion – we were previously able to observe an increase in the membrane thickness of polymersomes (from average 5.3 nm to 5.8 nm) and hybrids (from 4.9 to 5.3 nm), following the insertion of *bo₃* oxidase at relatively low concentration. With that notion in mind, we would expect to be able to detect changes in the thickness also in the present case, where SNAREs were reconstituted at much higher concentration.

8.) Paragraph on Kb measurements

The authors want to evaluate flexibility of the membrane through the measurement of Kb and see influence of membrane protein insertion on GUV. However the insertion of MP and tests of fusion has been made on LUV/SUV. Can the results be transposed to what occurs for LUV /SUV ? What about membrane curvature effect ? Please add comments on this part.

Values of Kb are extremely Low (4-6kT) even lower than values reported for classical liposomes membranes (~15kT). Apart from the eventual effect of dye, snare etc were such values expected compared to pure vesicle (pure polymersomes, pure soy PC ? what about literature ?)

The membrane bending rigidity is a material property and does not depend on the patch size of membrane, neither on how curved the membrane is.

Regarding the relatively low bending rigidity – very often inclusions lead to softening of the membrane (e.g., observed due to partitioning of co-surfactants and transmembrane inclusions; 10.1051/jphys:01986004703050700.jpa-00210231, 10.1103/PhysRevLett.76.4436, 10.1103/PhysRevE.67.012901). In the present work, soy PC was used to form lipid and hybrid vesicles. Soy PC is not a single pure lipid but a lipid mixture, exhibiting heterogeneous lipid and fatty acid distribution (<https://avantilipids.com/product/441601>). This composition has a similar effect to the aforementioned membrane inclusions and can result in softening of the membrane, in comparison to the frequently used liposomes made of a single lipid. Furthermore, softening of pure lipid

(doi:10.1088/1367-2630/13/2/025004) or polymer (10.1073/pnas.1919306117) membranes due to the insertion of very low concentration of peptides/MVs was reported previously.

9.) Paragraph :Fusion and light scattering experiment :

There is no comment about the smallest population in Hybrid vesicles (around 40nm which evolve to 60nm after fusion) and 24nm for pure polymersomes. As it is intensity based size distribution, this is the major population. Is it correlated with CryoTEM results ? It would be nice for cryo-TEM to add pictures with larger scale and precise size distribution by cryo-TEM (on a few hundred of vesicles)

We appreciate this suggestion – large scale micrographs as well as the precise size distribution of hybrids ($N = 674$) and polymersomes ($N = 1011$) determined with cryo-EM can be found in the newly added Fig. S1.

10.) Is there a role of the preparation process on the fusion of the vesicle ?

Pure polymersomes vesicle fuse progressively during time which is unexpected. Are the Polymersomes prepared by classical procedure (hydration extrusion process) fuse during time ? This should be checked.

The fusogenicity of proteovesicles can be significantly affected by the preparation process (direct reconstitution into preformed, partially solubilized vesicles vs. formation of proteovesicles from mixed micelles). With respect to the latter, poor fusogenicity of SNAREs-containing vesicles prepared via direct protein insertion is frequently reported in the works by Rizo and colleagues, while high activity can be seen in proteovesicles prepared with the co-micellization method (see for instance 10.1529/biophysj.105.071415). This difference is broadly observed and recognized by the SNAREs community, although the underlying mechanism is not entirely understood. To account for these differences, in the present work, we employed co-micellization method as a preferred method for the reconstitution of SNAREs. In any case, the entirety of fusion, from start to finish, is expected to be affected by the method of preparation rather than only a certain portion of it. Finally, it is worth noting that the membrane mixing signal in hybrids and polymersomes could be seen plateauing as well, although at much later times (11+ hrs; please see figure, below).

11.) Paragraph : Functional coupling as crucial determinant for full fusion :

Finally, the polymersome alone present higher ATP synthesis than the other systems : is it because this polymersome is particular in the sense that it forms very thin membrane similar to those of liposomes and that it is very flexible ?

Is there a specific role of hybrid polymer/lipid vesicle here ?

We believe that the similar membrane thickness, fluidity, bending rigidity and surface charge of natural and polymer-based membranes are crucial parameters governing the efficient and rapid SNARE-mediated fusion. Nevertheless, the importance of the low edge tension was highlighted by the functional coupling experiments, whereby the largest degree of content mixing was observed in the system with the most stable and persistent pores – polymersomes. It is important to note that the

fusion can in general be arrested in intermediate states (docking, hemifusion, transient pore opening), but in our setup, only full fusion (pore opening further progressing towards the merger of lumens) will result in functional coupling between bo_3 oxidase and ATP synthase. Therefore, it is of no surprise that the stable pore opening translates so well into high ATP synthesis.

As discussed in the previous points and as rationalized in the new segments of the Introduction and Conclusions sections, hybrid vesicles were not formed in order to influence fusion in any way. Instead, hybrid architecture is important and attractive because of the previously mentioned reasons (benefits for MP insertions, decreased permeability upon protein insertion, extended functional lifetime of systems fortified with hybrids, etc.), therefore our goal was to explore SNARE-mediated fusion of hybrids along with polymersomes (and use liposomes as a point of comparison/benchmark).

12.) Paragraph : Similar membrane composition but different fusion progression

It is mentioned that PDMS-*g*-PEO predominantly form monolayers, but it has been reported in previous studies on this copolymer, through Langmuir isotherm experiments and Static light scattering studies that this copolymer do form membrane with a bilayer (hairpin conformation of the chains).³ The authors should precise the reference where this information about predominant monolayer is available for this copolymer and/or discuss this section in regard of the paper aforementioned. May be mechanism of hemifusion can be applied to this copolymer.

As suggested, the reference (10.1021/nn4039589) was added to the statement made in the second paragraph of the section "Similar membrane composition but different fusion progression" to support the referenced claim. Under the conditions applied in our study, PDMS-*g*-PEO bilayers were not observed (PDMS-*g*-PEO was not subjected to osmotic stress, so the vesicles did not undergo transformation towards stomatocytes), which is consistent with the conclusion of the study by Salva and colleagues. It is important to note that the hairpin conformation referenced by the Reviewer 1 relates to triblock copolymers (10.1021/nn4039589, Fig. 8B) and not to PDMS-*g*-PEO, while the hemifusion initiated through the stalk formation was instead described with respect to diblock copolymers (10.1021/nn4039589, Fig. 8A). Furthermore, in the latter case, the term hemifusion is not synonymous with the same term used in the context of lipid bilayer fusion, since the polymer fusion event is limited to two monolayers instead of four, and since the pore opening supersedes stalk expansion directly. That is why we were careful to discriminate between the two instances of hemifusion when describing fusion progression in polymersomes and hybrids.

13.) Conclusion

In the conclusion the authors present the PDMS-*g*-PEO as ideal copolymer because of its high fluidity compared to the commonly used rigid diblock PBut-PEO. The copolymers PBut-*b*-PEO are not the most commonly used for membrane protein insertion but triblock PMOXA-PDMS-PMOXA or diblock PDMS-*b*-PMOXA are (See work of Palivan, Meier and col. for instance)⁴⁻⁷. Finally the authors should better precise in the conclusion what are the advantages/disadvantages of the particular grafted copolymer PDMS-*g*-PEO (apart to be commercially available). As aforementioned this system is relatively permeable, therefore would not be an ideal candidate in the development of nanoreactor/artificial cell well active control of diffusion of species is mandatory with low contribution of passive diffusion.

In the Conclusions, we summarize and highlight those characteristics of PDMS-*g*-PEO, which make this polymer a very promising candidate for the insertion of membrane proteins and, further on, the construction of multi-enzyme systems via SNARE-mediated fusion. We ascribe our claims to a variety of factors, some of which are explored in the present manuscript or and some in our previous works, and conclude that PDMS-*g*-PEO is an attractive alternative to commonly used lipids. We acknowledge the Reviewer's remark that the manuscript would be enriched by the discussion on other commonly used polymers, so we considerably extended this segment. Finally, in regards to PDMS-*g*-PEO permeability – the latter was determined to be comparable to the one measured in liposomes comprised of commonly used lipids, such as POPC, therefore we believe that the functionality of

various nanoreactors/artificial cell models would be supported by this membrane architecture (which we also showcased via the coupling between respiratory enzymes, where low proton permeability of the membrane is a crucial parameter governing ATP synthesis).

(1) Battaglia, G.; Ryan, A. J.; Tomas, S. Polymeric vesicle permeability: A facile chemical assay *Langmuir* 2006, 22, 4910.

(2) Carlsen, A.; Glaser, N.; Le Meins, J.-F.; Lecommandoux, S. Block Copolymer Vesicle Permeability Measured by Osmotic Swelling and Shrinking *Langmuir* 2011, 27, 4884.

(3) Dao, T. P. T.; Brûlet, A.; Fernandes, F.; Er-Rafik, M.; Ferji, K.; Schweins, R.; Chapel, J. P.; Fedorov, A.; Schmutz, M.; Prieto, M.; Sandre, O.; Le Meins, J. F. Mixing Block Copolymers with Phospholipids at the Nanoscale: From Hybrid Polymer/Lipid Wormlike Micelles to Vesicles Presenting Lipid Nanodomains *Langmuir* 2017, 33, 1705.

(4) Garni, M.; Einfalt, T.; Lomora, M.; Car, A.; Meier, W.; Palivan, C. G. Artificial Organelles: Reactions inside Protein-Polymer Supramolecular Assemblies *Chimia (Aarau)* 2016, 70, 424.

(5) Kowal, J. Ł.; Kowal, J. K.; Wu, D.; Stahlberg, H.; Palivan, C. G.; Meier, W. P. Functional surface engineering by nucleotide-modulated potassium channel insertion into polymer membranes attached to solid supports *Biomaterials* 2014, 35, 7286.

(6) Tanner, P.; Baumann, P.; Enea, R.; Onaca, O.; Palivan, C.; Meier, W. Polymeric Vesicles: From Drug Carriers to Nanoreactors and Artificial Organelles *Accounts of Chemical Research* 2011, 44, 1039.

(7) Kumar, M.; Grzelakowski, M.; Zilles, J.; Clark, M.; Meier, W. Highly permeable polymeric membranes based on the incorporation of the functional water channel protein Aquaporin *Z Proc. Natl. Acad. Sci.* 2007, 104, 20719.

We appreciate the suggested literature, which was used to fortify the discussion.

REVIEWER 2

In the NCOMMS-21-06880-T manuscript, the authors presented the reconstitution fusogenic membrane proteins (SNAREs) within three different systems: (1) polymersomes, (2) the benchmarked liposomes and (3) a lipid/polymer hybrid vesicle. The authors reported that the inserted SNAREs proteins were functional by exhibiting the proper orientation in all their systems. Moreover, the authors proposed a mechanism of inter-vesicle fusion assisted by the SNAREs. The authors stipulated that both the presence of SNARE, and the physicochemical properties of polymersomes and hybrid vesicles (i.e. low bending rigidity and capacity to sustain long lasting pores) empower the rapid mixing of both their content and membrane. Taken together, the authors presented production of ATP by recreating a minimal respiratory chain achieved through the fusion of two distinct vesicles population, each bearing one of the functional membrane proteins for ATP production. Once mixed, and fused, ATP was produced. The paper is elegant and well written. The generation and analysis of protein-mediated fusion of polymer-based synthetic cells is novel and of particular importance for synthetic biology society. Therefore I would recommend this manuscript for publication following minor revision.

1.) First, the authors should clearly state what each type of error bars represent (s.e.m from what I believe?) within the caption of their figure. Even though the data distribution is shown, the clear indication of their nature is needed.

The requested details are now included in the figure captions.

2.) Moreover, statistical analysis between vesicle types/experimental conditions should be perform in order to justify their claims about improved functionality. Statistical analysis should also be explicitly stated in the caption (i.e. Fig 1D in the main text).

The analysis of content mixing efficiency in the absence and presence of SNAREs was added to Fig. 4, panel B. Furthermore, details of the analysis were added to the figure captions of Fig. 1 and Fig. 4, as

well as to the Materials and Methods, sections “SNARE-mediated content mixing via coupling of respiratory enzymes” and “Bending rigidity determination with the flickering spectroscopy and related scale-up of nano-sized vesicles to micron range”.

3.) In Fig. S7B, the authors should have the same units, and all the same conditions for the SNAREs-polymersomes in order to properly compare the two systems (i.e. ATP (nM) or Luminescence and w/ SNARE and w/o SNARE in all cases). Moreover, such experiment should be performed also on the benchmarked liposomes in order to gain additional insight on the performance of their novel systems. The y-axes of Fig. S7 (now Fig. S10) were unified. Additionally, the revised figure now includes comparison between the SNARE-free and SNARE-mediated content mixing of lipid, polymer and hybrid vesicles, as requested.

REVIEWER 3

In their manuscript, Otrin et al. report the fusion of polymer-based membranes induced by the fusogenic SNARE proteins. The authors show that SNARE proteins can be incorporated with good orientation in polymer and hybrid lipid/polymer membranes, characterize the bending rigidity and pore edge tension of the resulting membranes, and devise a plausible fusion mechanism based on the observation of key fusion intermediates by cryo-electron microscopy.

This is to the best of my knowledge the first report showing that SNARE proteins can induce fusion of polymer-based membranes, which makes it an important contribution to the field of synthetic cells. Extending the use of natural fusogenic proteins to artificial and hybrid membranes opens indeed perspectives for the reconstitution of life-like processes in well-defined artificial systems. Therefore, I recommend publication in Nature Communications. However, a number of critical points should be addressed before publication.

Main comments

1/ As a general comment, I found that the manuscript lacks clarity in some places, mainly because the order in which results are introduced is sometimes confusing.

* In the first paragraph of the results section, the authors write that “SNARE-functionalized nano-sized proteovesicles were spontaneously formed upon detergent removal”, but no data is shown here to back-up this claim. The authors should already refer here to DLS studies and cryo-EM images to back up the fact that nano-sized proteovesicles were indeed formed (these results are only shown later in the manuscript).

Segment of the manuscript mentioned by Reviewer 3 is now referring the reader to a new figure added to the supplemental information (Fig. S1), which depicts the size distribution of the formed vesicles, as determined via cryo-EM. Furthermore, large-scale micrographs were included in this figure in order to familiarize the reader with tested vesicles from the very start.

* The section about membrane mixing is quite confusing to me. I would suggest rephrasing/re-organizing most of this section. Results are a bit mixed up: for instance, the authors first discuss the absence of signal saturation in hybrids and polymersomes, then they switch to DLS results, then discuss the kinetics of membrane mixing, then go back to discussing the saturation in liposomes... Please re-organize in a more logical way. Similarly, I don't get the logic in the first paragraph with the sentences “Thereby, KCl neutralized the additional surface charge ...” and then “In addition, we omitted SNAREs from the vesicles...”. This is confusing (e.g., should “In addition” actually be “In control experiments...”). I would remove the comment “which should be valid for the polymer systems as well but at later times” or provide data at longer times to back up this point. Last, I am not sure why the variability in membrane mixing is shown as an inset in Figure 2A.

I think this should rather be shown in Figure 2D. Also, could the authors provide variability data on membrane mixing for the other two systems (liposomes and hybrids)?

Following the suggestions by Reviewer 3, the membrane mixing segment was rephrased to improve the coherence and cohesion. Furthermore, the inset of Fig. 2 was moved from panel A to panel D and is now shown next to the relevant data. Finally, we included information on the variability of membrane mixing in polymersomes to give a better impression about this phenomenon. The latter membrane mixing variability was found to be nearly the same to the one previously published for liposomes, as discussed in the manuscript. Furthermore, similar trend was observed in hybrid vesicles (a couple of measurements exhibiting the lowest and highest membrane mixing in this system are depicted in Fig. 2C). With this data, we aim to compare membrane mixing in natural and man-made membranes, while we are looking to systematically explore the intricacies of SNARE-mediated fusion of polymer-based membranes in our future works.

* In the content mixing experiment, discussion of the results shown in Fig. S7 should be more accurate. See in particular the sentence “As a result, we observed successful ATP production in both SNARE-functionalized and SNARE-free vesicles, although synthesis in the latter subsided rapidly (Fig. S7A), while the SNARE-mediated fusion resulted in a steady rate over more than an hour (Fig. S7B).” Actually, Fig. S7A shows results for hybrid vesicles, while Fig. S7B shows results for polymersomes, so the two figures are about two different systems. This should be explicit in the sentence. Is there any evidence that SNARE-mediated fusion also gives a steady rate over more than an hour for the other systems (liposomes and hybrids)? Could the authors also provide representative examples of the time-dependent ATP synthesis (as shown in SI Figure S7 for hybrids) for all three systems, with and without SNAREs? Also, why the y-axis on Fig. S7 has not been converted to ATP concentration as in Fig. S7A?

The data on SNARE-free and SNARE-mediated content mixing in liposomes, hybrids and polymersomes is now included in the revised Fig. S10, as suggested by Reviewer 2 as well as Reviewer 3. Moreover, the main text was rephrased to more accurately reflect the results presented in Fig. S10 (“Functional coupling as a crucial determinant for full fusion”, second paragraph).

* Last comment about clarity issues: in a few places the authors compare their current results to previously published studies, without mentioning the values they obtained in these previous studies. For instance, they write: “compared to vesicles formed in sucrose solution with low buffering capacity, the polymer swelled by nearly 20%, while the thickness of lipid bilayers remained unchanged”. It may be clearer to quote these precedent values from ref. 10. Similarly, they write “we saw nearly 50% decrease of the bending rigidity in protein-free polymersomes and hybrids formed in KCl, compared to the ones grown in sucrose.” Again, hard to judge if this assertion is valid without reading the cited reference. Please provide any relevant value directly in the manuscript. Also, were studies in reference 10 performed on the same vesicle systems?

We appreciate this suggestion – relevant values are now provided throughout the manuscript, where they were previously omitted. With respect to the last question, yes, the referenced studies were performed on the same systems, as described in the Materials and Methods section.

2/ Data should be added to back up the protein orientation studies. In particular, the corresponding SDS-PAGE gels should be shown in the SI. Also, there are no error bars on Fig. 1A, so it is unclear what the statistical relevance of these results is. Same comment about error bars applies to Fig. 1B.

New figures (Fig. S3 and Fig. S4) showing SDS-PAGE analysis of the SNAREs orientation and insertion efficiency were added to supplemental information. They include the requested pictures of gels, along with the measured band intensities. Furthermore, the experiments depicted in Fig. 1, panels A and B were reproduced and the newly collected data can be found in the revised Fig. 1. Description was added to the caption of Fig. 1 to reflect these changes.

3/ When the authors report membrane thickness values measured on cryo-EM images, what is the statistical relevance of the reported values? E.g., how many proteovesicles were analysed? Please provide any relevant statistics.

Analysed were 578 polymersomes as well as 976 polymer domains and 171 lipid domains in hybrid vesicles. The relevant statistics was added to the main text (“Correct orientation of SNAREs and low bending rigidity of the synthetic membranes as prerequisites for fusion”, second paragraph).

4/ Why are there two vesicle populations on DLS measurements (Fig. 2E-G)?

Vesicles formed from mixed micelles via detergent removal by size exclusion chromatography eluted as two distinct populations. We believe this to be due to localized differences in the speed of detergent removal in the column (faster detergent removal generally results in the formation of smaller vesicles, while gradual detergent removal favours the formation of larger vesicles). Since intensity-based size distribution is depicted in Fig. 2E-G, the larger vesicles appear more numerous than they actually are (discussion on this topic can be found in various places, for instance please see: <https://www.materials-talks.com/wp-content/uploads/2017/01/FAQ-Calculating-volume-distributions-from-DLS-data.pdf>). To give the reader a better impression of the relationship between the size data from DLS and cryo-EM, Fig. S1 was added.

Other comments

5/ In the introduction, I reckon studies from Lecommandoux’s and Palivan’s groups on the construction of polymer vesicles as synthetic cells should be cited.

The Introduction section was expanded considerably – please see Reviewer 1, Points 1, 2, 12 and 13 for details.

6/ I would suggest plotting Figs. 2B-D with the same scale on the y-axis, this would make it easier to compare the data. Same comment for Figs. 2E-G regarding the x-axis: easier to compare if the same scale on the x-axis is used (but the scale on the y-axis can be kept as it is now).

Mentioned figures were replotted, as suggested.

7/ Please define what CM stands for in SI Figure S6 (I guess it stands for “content mixing”?).

Reviewer 3 is correct in their assumption. “CM” was defined in figure caption to avoid ambiguity.

8/ The overall manuscript is a bit long (and in particular the conclusion), so might be worth condensing it a bit.

The manuscript was condensed where possible to the best of our abilities.

REVIEWER COMMENTS

Reviewer #1 (Remarks to the Author):

Comment on the revised version and rebuttal letter of the paper

« *En route to dynamic life processes by SNARE-mediated fusion of polymer and hybrid membranes* »

Most of the remarks and questions have been discussed and commented by the authors and the manuscript quality has been really improved
However I have a last remark about a question I asked : I still disagree with the authors , and to my opinion this point needs clarification before publication

See below :

12.) Paragraph : Similar membrane composition but different fusion progression
It is mentioned that PDMS-g-PEO predominantly form monolayers , but it has been reported in previous studies on this copolymer, through Langmuir isotherm experiments and Static light scattering studies that this copolymer do form membrane with a bilayer (hairpin conformation of the chains).³ The authors should precise the reference where this information about predominant monolayer is available for this copolymer and/or discuss this section in regard of the paper aforementioned. May be mechanism of hemifusion can be applied to this copolymer.

As suggested, the reference (10.1021/nn4039589) was added to the statement made in the second paragraph of the section “Similar membrane composition but different fusion progression” to support the referenced claim. Under the conditions applied in our study, PDMS-g-PEO bilayers were not observed (PDMS-g-PEO was not subjected to osmotic stress, so the vesicles did not undergo transformation towards stomatocytes), which is consistent with the conclusion of the study by Salva and colleagues. It is important to note that the hairpin conformation referenced by the Reviewer 1 relates to triblock copolymers (10.1021/nn4039589, Fig. 8B) and not to PDMS-g-PEO, while the hemifusion initiated through the stalk formation was instead described with respect to diblock copolymers (10.1021/nn4039589, Fig. 8A). Furthermore, in the latter case, the term hemifusion is not synonymous with the same term used in the context of lipid bilayer fusion, since the polymer fusion event is limited to two monolayers instead of four, and since the pore opening supersedes stalk expansion directly. That is why we were careful to discriminate between the two instances of hemifusion when describing fusion progression in polymersomes and hybrids.

I am sorry to confirm and maintain that the reference that I mentioned (Dao et al. Langmuir 2017) DOES CONSIDER PDMS-g-PEO copolymer (see the supporting information of the article) although of course, the article mainly deals with triblock copolymers.

Moreover in the reference added (salva et al.), the authors themselves suggested that membrane of PDMS-g-PEO probably looks like a classic lipid bilayer. (see indeed the figure caption of Figure 8)

I do not agree with the comment added « PDMS-g-PEO forms predominantly monolayers at moderate osmotic conditions », referring to salva et al. Salva and al. do not show that. Their results instead, by the obtention of nested vesicle with PDMS-g-PEO vesicle in hypertonic conditions suggest a bilayer conformation of PDMS-g-PEO.

This was later confirmed by the Dao et al with quantitative data. (Dao et al. Langmuir 2017)

A part from the complex mechanism of fusion, my message is that the reader must be aware that membrane of PDMS-g-PEO is constituted of a bilayer. Quantitative data support it.

To conclude, this paragraph has to be reconsidered. The mechanism of fusion has probably lot of similitudes with lipidic bilayers.

Reviewer #2 (Remarks to the Author):

The authors answered all my concerns/comments. Therefore, I would like to recommend the manuscript for publication in the current form.

Reviewer #3 (Remarks to the Author):

The authors have addressed all my comments in this revised version. They have done changes to the main text and have included additional analyses/experiments that I think strengthen the conclusions of the work. Of particular relevance to my concerns, the state of the art has been considerably expanded, the clarity of the main text has been improved, and data has been added to confirm protein orientation in the vesicles.

Based on these observations, the overall quality of the manuscript has improved and I now support publication in Nature Communications.

We would like to thank the Reviewers for carefully reassessing the edited manuscript. We are very happy to hear that all but one of the points that they raised were addressed successfully. Included below is a detailed response to the last remaining open question, whereby comments by Reviewer 1 are written in black, our previous response in blue and the current one in purple.

Paragraph : Similar membrane composition but different fusion progression It is mentioned that PDMS-g-PEO predominantly form monolayers, but it has been reported in previous studies on this copolymer, through Langmuir isotherm experiments and Static light scattering studies that this copolymer do form membrane with a bilayer (hairpin conformation of the chains).³ The authors should precise the reference where this information about predominant monolayer is available for this copolymer and/or discuss this section in regard of the paper aforementioned. May be mechanism of hemifusion can be applied to this copolymer.

As suggested, the reference (10.1021/nn4039589) was added to the statement made in the second paragraph of the section “Similar membrane composition but different fusion progression” to support the referenced claim. Under the conditions applied in our study, PDMS-g-PEO bilayers were not observed (PDMS-g-PEO was not subjected to osmotic stress, so the vesicles did not undergo transformation towards stomatocytes), which is consistent with the conclusion of the study by Salva and colleagues. It is important to note that the hairpin conformation referenced by the Reviewer 1 relates to triblock copolymers (10.1021/nn4039589, Fig. 8B) and not to PDMS-g-PEO, while the hemifusion initiated through the stalk formation was instead described with respect to diblock copolymers (10.1021/nn4039589, Fig. 8A). Furthermore, in the latter case, the term hemifusion is not synonymous with the same term used in the context of lipid bilayer fusion, since the polymer fusion event is limited to two monolayers instead of four, and since the pore opening supersedes stalk expansion directly. That is why we were careful to discriminate between the two instances of hemifusion when describing fusion progression in polymersomes and hybrids.

1.) I am sorry to confirm and maintain that the reference that I mentioned (Dao et al. Langmuir 2017) DOES CONSIDER PDMS-g-PEO copolymer (see the supporting information of the article) although of course, the article mainly deals with triblock copolymers.

We have once again carefully analysed the initially suggested reference (Dao et al., Langmuir 2017) which focuses on hybrid (lipid/polymer mixture) architectures and related phase separation. As mentioned by Reviewer 1, indeed, some data on PDMS-g-PEO hybrids is included in the Supporting information; we never disputed that. Nevertheless, said data is not in agreement with statements made by the Reviewer. For instance, the membrane thickness of PDMS-g-PEO polymersomes as determined by cryo-EM was reported to be around 5.5 nm (SI, page 26, Fig. S31), which is a thickness associated with PDMS-g-PEO monolayers (Salva et al., 2013, Table 2: the reported determined thickness of unilamellar vesicles was 5.4 ± 0.6 nm). Next, let us consider the data associated with

PDMS-g-PEO/DPPC hybrid vesicles, even though this hybrid blend is not directly related to our study. First, the morphologies of PDMS-g-PEO/DPPC hybrids (85/15 %) were analysed by the authors. The results of the analysis summarized on page 32, Fig. S47 clearly show that the occurrence of nested vesicles (ones exhibiting bilayers) was negligible (less than 5 % of all vesicles, while the rest were reported to be unilamellar). Morphology data is accompanied with size distribution one, which indicates a membrane thickness of approx. 6 nm, again, characteristic for polymer monolayers. Similar trend could be seen with respect to another tested composition of hybrid vesicles (79/21%, polymer/lipid), whereby the recorded membrane thickness was about 5.8 nm (SI, page 33, Fig. S50) and the authors report on not a single nested vesicle (SI, page 33, Fig. S49). All findings mentioned above are also conveniently compiled in SI, page 34, Table S10. A single glance at this table would reveal that no multilamellar vesicles were discovered by Dao et al. (Table S10, 8th column).

Moreover, we believe that our answer to the original comment “It is important to note that the hairpin conformation referenced by the Reviewer 1 relates to triblock copolymers (10.1021/nn4039589, Fig. 8B) and not to PDMS-g-PEO, ...” has been misunderstood. Salva et al. and Dao et al. do definitely consider PDMS-g-PEO and we never disputed that. However, the hairpin conformation is to the best of our knowledge predominantly associated with triblock copolymers (e.g., 10.1021/nl051515x) and the same hairpin conformation has been discussed in Salva et al., exclusively with respect to PEO-b-PDMS-b-PEO in an effort to qualitatively interpret its resistance to fusion (and bilayer formation), maintaining the stomatocyte shape.

2.) *Moreover in the reference added (salva et al.), the authors themselves suggested that membrane of PDMS-g-PEO probably looks like a classic lipid bilayer. (see indeed the figure caption of Figure 8)*

We would encourage Reviewer 1 to read the rest of the manuscript by Salva et al., so to provide correct context. The authors clearly state (page 4, 2nd paragraph): *»Figure 2A shows typical cryo-TEM image of PDMS-g-PEO vesicles in their native state (isotonic conditions, i.e., pure water). ... A statistical study (over a number of structures, n = 148) revealed that 96% of the objects were unilamellar vesicles. Indeed, the difference between simple (unilamellar) and double (bilamellar) membranes is clear on cryo-TEM images since silicon atoms of PDMS provide an electron scattering density sufficiently different from the carbon and oxygen atoms of PEO.«* Next, the authors subjected these vesicles to hypertonic stress, thereby deflating them (Fig. 4). The rapid uncompensated increase in membrane surface was reported to lead into vesicle shape transformation from spherical to stomatocytes. Due to the high fluidity of PDMS-g-PEO, the stoma (mouth hole) of the latter was seen to reseal via hemifusion of proximal monolayers, resulting in the formation of nested vesicles (double layered). Nested vesicles are easily distinguishable by cryo-EM due to their characteristic appearance (Fig. 2) as well as their significantly increased membrane thickness (from 5.4 to 15.5 nm, Table 2, 2nd and 3rd row). In summary, yes, bilayers were observed by Salva et al. **but only under hypertonic conditions, while only monolayers could be seen under isosmotic conditions.**

With this data in mind, one can draw parallels with our study. All types of vesicles were prepared and evaluated under isosmotic conditions. Unsurprisingly, no polymer bilayers were observed by either cryo-EM (measured membrane thickness was 6.1 to 6.6 nm and no bilayers could be seen on the recorded micrographs) or content mixing experiments (a thickness of about 16 nm corresponding to bilayer would create a large hydrophobic mismatch between the polymer and the lipids or membrane proteins, which in turn would favour phase separation or decreased enzyme activity, respectively). Finally, no polymer bilayers were observed also in the context of our recent study (Marusic et al., 2020). These previously pointed out findings remain entirely overlooked by Reviewer 1.

3.) *I do not agree with the comment added « PDMS-g-PEO forms predominantly monolayers at moderate osmotic conditions », referring to salva et al. Salva and al. do not show that.*

Salva et al. not only show this but also explicitly state it – please see the text cited in Point 2, in bold. Nevertheless, we rephrased this statement to avoid any possible misinterpretation by the readers. The statement now reads as:

*» ... although PDMS-g-PEO forms predominantly monolayers **under isosmotic conditions (applied also in this study)**⁵¹ and therefore hemifusion loses its original meaning.«*

4.) *Their results instead, by the obtention of nested vesicle with PDMS-g-PEO vesicle in hypertonic conditions suggest a bilayer conformation of PDMS-g-PEO.*

Indeed, under hypertonic conditions. Meanwhile, in current study, vesicles were instead prepared and evaluated under isosmotic condition, as stated above.

5.) *This was later confirmed by the Dao et al with quantitative data. (Dao et al. Langmuir 2017)*
Data provided by Dao et al. does not support these claims, as discussed in Point 1.

6.) *A part from the complex mechanism of fusion, my message is that the reader must be aware that membrane of PDMS-g-PEO is constituted of a bilayer. Quantitative data support it. To conclude, this paragraph has to be reconsidered.*

Yes, under specified conditions not relevant to the present study, as discussed above. Nevertheless, our statement was rephrased to avoid any potential confusion.

7.) *The mechanism of fusion has probably lot of similitudes with lipidic bilayers.*

Overall, we hope that the reviewer and potential readers will appreciate that we are intentionally looking for biomimetic properties in line with their comment »The mechanism of fusion has probably lot of similitudes with lipidic bilayers.« and that the fact that SNARE-mediated fusion is attainable with the PDMS-g-PEO vesicles represents such a similarity itself.

See my last remarks in red.

We would like to thank the Reviewers for carefully reassessing the edited manuscript. We are very happy to hear that all but one of the points that they raised were addressed successfully. Included below is a detailed response to the last remaining open question, whereby comments by Reviewer 1 are written in black, our previous response in blue and the current one in purple.

Paragraph : Similar membrane composition but different fusion progression It is mentioned that PDMS-*g*-PEO predominantly form monolayers , but it has been reported in previous studies on this copolymer, through Langmuir isotherm experiments and Static light scattering studies that this copolymer do form membrane with a bilayer (hairpin conformation of the chains).³ The authors should precise the reference where this information about predominant monolayer is available for this copolymer and/or discuss this section in regard of the paper aforementioned. May be mechanism of hemifusion can be applied to this copolymer.

As suggested, the reference (10.1021/nn4039589) was added to the statement made in the second

paragraph of the section “Similar membrane composition but different fusion progression” to support the referenced claim. Under the conditions applied in our study, PDMS-*g*-PEO bilayers were not observed (PDMS-*g*-PEO was not subjected to osmotic stress, so the vesicles did not undergo transformation towards stomatocytes), which is consistent with the conclusion of the study by Salva and colleagues. It is important to note that the hairpin conformation referenced by the Reviewer 1 relates to triblock copolymers (10.1021/nn4039589, Fig. 8B) and not to PDMS-*g*-PEO, while the hemifusion initiated through the stalk formation was instead described with respect to diblock copolymers (10.1021/nn4039589, Fig. 8A). Furthermore, in the latter case, the term hemifusion is not synonymous with the same term used in the context of lipid bilayer fusion, since the polymer fusion event is limited to two monolayers instead of four, and since the pore opening supersedes stalk expansion directly. That is why we were careful to discriminate between the two instances of hemifusion when describing fusion progression in polymersomes and hybrids.

1.) I am sorry to confirm and maintain that the reference that I mentioned (Dao et al. Langmuir 2017)

DOES CONSIDER PDMS-*g*-PEO copolymer (see the supporting information of the article) although of course, the article mainly deals with triblock copolymers.

We have once again carefully analysed the initially suggested reference (Dao et al., Langmuir 2017) which focuses on hybrid (lipid/polymer mixture) architectures and related phase separation. As

Self-assembled vesicles of amphiphili

mentioned by Reviewer 1, indeed, some data on PDMS-*g*-PEO hybrids is included in the Supporting

information; we never disputed that. Nevertheless, said data is not in agreement with statements made by the Reviewer. For instance, the membrane thickness of PDMS-*g*-PEO polymersomes as determined by cryo-EM was reported to be around 5.5 nm (SI, page 26, Fig. S31), which is a thickness associated with PDMS-*g*-PEO monolayers (Salva et al., 2013, Table 2: the reported determined thickness of unilamellar vesicles was 5.4 ± 0.6 nm).

Next, let us consider the data associated with PDMS-*g*-PEO/DPPC hybrid vesicles, even though this hybrid blend is not directly related to our study.

First, the morphologies of PDMS-*g*-PEO/DPPC hybrids (85/15 %) were analysed by the authors. The results of the analysis summarized on page 32, Fig. S47 clearly show that the occurrence of nested vesicles (ones exhibiting bilayers) was negligible (less than 5 % of all vesicles, while the rest were reported to be unilamellar). Morphology data is accompanied with size distribution one, which indicates a membrane thickness of approx. 6 nm, again, characteristic for polymer monolayers. Similar trend could be seen with respect to another tested composition of hybrid vesicles (79/21%, polymer/lipid), whereby the recorded membrane thickness was about 5.8 nm (Si, page 33, Fig. S50)

and the authors report on not a single nested vesicle (SI, page 33, Fig. S49). All findings mentioned above are also conveniently compiled in SI, page 34, Table S10. A single glance at this table would reveal that no multilamellar vesicles were discovered by Dao et al. (Table S10, 8th column). Moreover, we believe that our answer to the original comment “It is important to note that the hairpin conformation referenced by the Reviewer 1 relates to triblock copolymers (10.1021/nn4039589, Fig. 8B) and not to PDMS-g-PEO, ...” has been misunderstood. Salva et al. and Dao et al. do definitely consider PDMS-g-PEO and we never disputed that. However, the hairpin conformation is to the best of our knowledge predominantly associated with triblock copolymers (e.g., 10.1021/nl051515x) and the same hairpin conformation has been discussed in Salva et al., exclusively with respect to PEO-b-PDMS-b-PEO in an effort to qualitatively interpret its resistance to fusion (and bilayer formation), maintaining the stomatocyte shape.

10.1021/nl051515x : in this very interesting work from Klein and col. it has been indeed shown in simulation of tubular structure (not vesicles). Actually the proportion of loop (hairpin) and bridge conformation of triblock copolymers in hybrid vesicle depend on the hydrophobic block length (see nice recent study from Tsao and col. [10.3390/polym12030639](https://doi.org/10.3390/polym12030639))

2.) Moreover in the reference added (salva et al.), the authors themselves suggested that membrane of PDMS-g-PEO probably looks like a classic lipid bilayer. (see indeed the figure caption of Figure 8)

We would encourage Reviewer 1 to read the rest of the manuscript by Salva et al., so to provide correct context. The authors clearly state (page 4, 2nd paragraph): »Figure 2A shows typical cryo-TEM image of PDMS-g-PEO vesicles in their native state (**isotonic conditions**, i.e., pure water). ... A

statistical study (over a number of structures, $n = 148$) **revealed that 96% of the objects were unilamellar vesicles. Indeed, the difference between simple (unilamellar) and double (bilamellar) membranes is clear on cryo-TEM images since silicon atoms of PDMS provide an electron scattering density sufficiently different from the carbon and oxygen atoms of PEO.**

« Next, the authors subjected these vesicles to hypertonic stress, thereby deflating them (Fig. 4). The rapid uncompensated increase in membrane surface was reported to lead into vesicle shape transformation from spherical to stomatocytes. Due to the high fluidity of PDMS-g-PEO, the stoma (mouth hole) of the latter was seen to reseal via hemifusion of proximal monolayers, resulting in the formation of nested vesicles (double layered). Nested vesicles are easily distinguishable by cryo-EM due to their characteristic appearance (Fig. 2) as well as their significantly increased membrane thickness (from 5.4 to 15.5 nm, Table 2, 2nd and 3rd row). In summary, yes, bilayers were observed by Salva et al. **but only under hypertonic conditions, while only monolayers could be seen under isosmotic conditions.**

With this data in mind, one can draw parallels with our study. All types of vesicles were prepared and evaluated under isosmotic conditions. Unsurprisingly, no polymer bilayers were observed by either cryo-EM (measured membrane thickness was 6.1 to 6.6 nm and no bilayers could be seen on the recorded micrographs) or content mixing experiments (a thickness of about 16 nm corresponding to bilayer would create a large hydrophobic mismatch between the polymer and the lipids or membrane proteins, which in turn would favour phase separation or decreased enzyme activity, respectively). Finally, no polymer bilayers were observed also in the context of our recent study (Marusic et al., 2020). These previously pointed out findings remain entirely overlooked by Reviewer 1.

Thanks for your advice to read carefully the paper from Salva et al. but I am really, really aware of its content...

In the Figure 2 A of this article, indeed in pure water, and yes 96% of vesicles are unilamellar etc..etc..

Anyway, finally I wonder if there is a simple problem of misunderstanding between us. what do you call monolayer ? the fact that in cryoTEM we see that ?

-Figure 1 (called in the article unilamellar in the work of Salva et al.)

And bilayer that ?

(Figure 2 called in the article bilamellar or nested vesicle in the work of Salva et al.)

If so, I completely understand your comment « PDMS-g-PEO forms predominantly monolayers at Moderate osmotic conditions »

I think that you are talking about the number of membrane layer (unilamellar : monolayer, bilamellar(nested) : bilayer etc...) which indeed vary with the osmotic pressure.

But unilamellar do not imply systematically a monolayer (for instance, when we are talking of liposomes GUV for instance, Giant **Unilamellar** vesicles, the membrane is constituted of a **bilayer** of phospholipids)

I was talking about the conformation of the polymer chain *inside* the membrane PDMS-g-PEO does form, in a membrane like figure 1, a bilayer (with hairpin conformation)

to be as clear as possible, I have drawn a quick diagram

Figure 3

This PDMS-g-PEO chain do present an hairpin conformation in the membrane, as it was suggested in the work of Salva et al, (that's why, in hypertonic condition, by adding glucose solution on vesicle prepared initially in pure water 'so isotonic condition, you can obtained nested vesicles (what you call bilayer) And that why for triblock copolymer, as it was suggested by the authors (but no real proof, that's true) you can not reach nested vesicle (closed bilamellar structure)

Surprisingly, the small amount of vesicles which undergo invagination under hypertonic conditions (6.4% of population) never reached a closed bilamellar structure and kept their stomatocyte shape, even though the R_0 determined for most of them was higher than $R_0^{250 \text{ mM}}$. A probable explanation of the phenomenon lies in the triblock architecture of the PEO-*b*-PDMS-*b*-PEO copolymer, which leads to the formation of a well-defined copolymer monolayer and not to a bilayer classically formed by diblock copolymers. This monolayer nanostructure disfavors the stomatocyte closure because all the PEO-*b*-PDMS-*b*-PEO chains involved in the hemifusion step would have to go through a very unfavorable hairpin conformation to avoid contact between the PDMS hydrophobic core and water, thus increasing the energetic cost of the membrane fusion (Figure 8).

In this paper also, Salva and al. talk about monolayer, or bilayer, to describe the conformation of the chain **inside** the membrane. (not the number of « membrane », for that they use the term unilamellar or bilamellar)

And I do confirm that the hairpin conformation (leading to a bilayer) of the PDMS-g-PEO chain inside a membrane in a polymerosome, has been proven by static light scattering and Langmuir isotherm experiments by Dao et al.

I hope that this will clarify my comments and the work of Salva et al. that I know very well... and will help you to understand all my other remarks below, that I did for the first reviewing round.

Finally, the slight modification of your text « although PDMS-g-PEO forms predominantly monolayers under isosmotic conditions (applied also in this study)⁵¹ » and therefore hemifusion loses its original meaning » is not sufficient and I think still based on a misunderstanding between us. Still, a non-negligible part of the future readers can think that PDMS-g-PEO chain present a monolayer (so in fact in their spirit an extended conformation of the chain in the membrane, not hairpin) which is not true.

This has to be clarified as it is very important, in order that future readers do not report in their future work, not intentionally, wrong information based on this reading. Please make a clear distinction between what you call monolayer and copolymer conformation in the membrane. Again a membrane of a simple vesicle is not systematically a monolayer, but this could be interpreted like this with your comment.

And I clearly think that what you observe is explained by the fact, that, **inside** a membrane, the grafted PDMS-g-PEO chain present an hairpin conformation.

3.) I do not agree with the comment added « PDMS-g-PEO forms predominantly monolayers at moderate osmotic conditions », referring to Salva et al. Salva and al. do not show that.

Salva et al. not only show this but also explicitly state it – please see the text cited in Point 2, in bold. Nevertheless, we rephrased this statement to avoid any possible misinterpretation by the readers. The statement now reads as:

» ... although PDMS-g-PEO forms predominantly monolayers **under isosmotic conditions (applied also in this study)**⁵¹ and therefore hemifusion loses its original meaning.«

See above

4.) *Their results instead, by the obtention of nested vesicle with PDMS-g-PEO vesicle in hypertonic conditions suggest a bilayer conformation of PDMS-g-PEO.*

Indeed, under hypertonic conditions. Meanwhile, in current study, vesicles were instead prepared and evaluated under isosmotic condition, as stated above.

See above

5.) *This was later confirmed by the Dao et al with quantitative data. (Dao et al. Langmuir 2017)*
Data provided by Dao et al. does not support these claims, as discussed in Point 1.

See above

6.) *A part from the complex mechanism of fusion, my message is that the reader must be aware that membrane of PDMS-g-PEO is constituted of a bilayer. Quantitative data support it. To conclude, this paragraph has to be reconsidered.*

Yes, under specified conditions not relevant to the present study, as discussed above. Nevertheless, our statement was rephrased to avoid any potential confusion.

See above

7.) *The mechanism of fusion has probably lot of similitudes with lipidic bilayers.*

Overall, we hope that the reviewer and potential readers will appreciate that we are intentionally looking for biomimetic properties in line with their comment »The mechanism of fusion has probably lot of similitudes with lipidic bilayers.« and that the fact that SNARE-mediated fusion is attainable with the PDMS-g-PEO vesicles represents such a similarity itself.

But I do appreciate this nice work, just clarify, please, this story of monolayer, bilayer and conformation of the chain inside the membrane.

Many thanks to Reviewer 1 (text in red) for an insightful discussion on the architecture of PDMS-g-PEO membranes, as well as for the provided supporting material. Our comments can be found below, in blue.

10.1021/nl051515x : in this very interesting work from Klein and col. it has been indeed shown in simulation of tubular structure (not vesicles). Actually the proportion of loop (hairpin) and bridge conformation of triblock copolymers in hybrid vesicle depend on the hydrophobic block length (see nicerecent study from Tsao and col. 10.3390/polym12030639)

Thanks for your advice to read carefully the paper from salva et al. but I am really, really aware of its content....

In the Figure 2 A of this article, indeed in pure water, and yes 96% of vesicles are unilamellar etc..etc..

Anyway, finally I wonder if there is a simple problem of misundersanting between us.what do you call monolayer ? the fact that in cryotem we see that ?

-Figure 1 (called in the article unilamellar in the work of Salva et al.)

And bilayer that ?

(Figure 2 called in the article bilamellar or nested vesicle in the work of Salva et al.)

If so, I completely understand your comment « *PDMS-g-PEO forms predominantly monolayers atModerate osmotic conditions* »

I think that your are talking about the number of membrane layer (unilamellar : monolayer,bilamellar(nested) : bilayer etc...) which indeed vary with the osmotic pressure.

But unilamellar do not imply systematically a monolayer (for instance, when we are talking of liposomes GUV for instance, Giant **Unilamellar** vesicles, the membrane is constituted of a **bilayer** of phospholipids)

I was talking about the conformation of the polymer chain *inside* the membrane PDMS-g-PEO does form, in a membrane like figure 1, a bilayer (with hairpin

conformation)to be as clear as possible, I have drawn a quick diagram

Figure 3

This PDMS-g-PEO chain do present an hairpin conformation in the membrane, as it was suggested in the work of Salva et al, (that's why, in hypertonic condition, by adding glucose solution on vesicle prepared initially in pure water 'so isotonic condition, you can obtained nested vesicles (what you call bilayer) And that why for triblock copolymer, as it was suggested by the authors (but no real proof, that's true) you can not reach nested vesicle (closed bilamellar structure)

Surprisingly, the small amount of vesicles which undergo invagination under hypertonic conditions (6.4% of population) never reached a closed bilamellar structure and kept their stomatocyte shape, even though the R_0 determined for most of them was higher than $R_0^{250 \text{ mM}}$. A probable explanation of the phenomenon lies in the triblock architecture of the PEO-*b*-PDMS-*b*-PEO copolymer, which leads to the formation of a well-defined copolymer monolayer and not to a bilayer classically formed by diblock copolymers. This monolayer nanostructure disfavors the stomatocyte closure because all the PEO-*b*-PDMS-*b*-PEO chains involved in the hemifusion step would have to go through a very unfavorable hairpin conformation to avoid contact between the PDMS hydrophobic core and water, thus increasing the energetic cost of the membrane fusion (Figure 8).

In this paper also, salva and al. talk about monolayer , or bilayer, to describe the conformation of the chain **inside** the membrane. (not the number of « membrane », for that they use the term unilamellar or bilamellar)

And I do confirm that the hairpin conformation (leading to a bilayer) of the PDMS-g-PEO chain inside a membrane in a polymerosme , has been proven by static light scattering and langmuir isotherm experiments by Dao et al.

I hope that this will clarify my comments and the work of Salva et al. that I know very well... and will help you to understand all my other remarks below, that I did for the first reviewing round.

Indeed, it would appear that there was a misunderstanding between us with respect to the usage of terminology. Thanks to the clarifications by Reviewer 1 it is now understood that they were referring to the intramembrane organization and not to the multimembrane stacks when discussing the potential of PDMS-g-PEO membranes for facilitating hemifusion.

Finally , the slight modification of your text « although PDMS-g-PEO forms predominantly monolayers under isosmotic conditions (applied also in this study)⁵¹ » and therefore hemifusion loses its original meaning » is not sufficient and I think still based on a misunderstanding between us. Still, a non negligible part of the future readers can think that PDMS-g-PEO chain present a monolayer (so in fact in their spirit a extended conformation of the chain in the membrane, not

hairpin) which is not true.

This has to be clarified as it is very important, in order that future readers do not report in their future work, not intentionally, wrong information based for instance on this reading. Please make a clear distinction between what you call monolayer and copolymer conformation in the membrane. Again a membrane of a simple vesicle is not systematically a monolayer, but this could be interpreted like this with your comment.

And I clearly think that what you observe is explained by the fact, that, **inside** a membrane, the grafted PDMS-g-PEO chain presents a hairpin conformation.

In light of the above-mentioned new understanding, we fully agree that the previous modification of our text is inaccurate. Strong evidence for a PDMS-g-PEO intramembrane organization as polymer bilayers can be found in works of Dao et al. (10.1021/acs.langmuir.6b04478) and Salva et al. (10.1021/nn4039589). Based on this information, we reworded the text to now read as:

“Considering the previously proposed organization of PDMS-g-PEO molecules (hairpin conformation) as a bilayer^{51,52}, we note that we adopted the terminology from lipid bilayers to provide an analogy, although some dissimilarities between the two systems cannot be excluded.”

Both mentioned works supporting this claim are cited in the above statement.

But I do appreciate this nice work, just clarify, please, this story of monolayer, bilayer and conformation of the chain inside the membrane.

We also appreciate a constructive discussion with Reviewer 1 and their kind final words. Many thanks also for taking the time to draw helpful sketches supporting their arguments.